# THE CURIOUS CASE OF BENIGN MEMORIZATION

**Sotiris Anagnostidis\*, Gregor Bachmann\*, Lorenzo Noci\*, Thomas Hofmann**
Department of Computer Science
ETH Zürich, Switzerland
`{sotirios.anagnostidis,gregor.bachmann,lorenzo.noci}@inf.ethz.ch`

## ABSTRACT

Despite the empirical advances of deep learning across a variety of learning tasks, our theoretical understanding of its success is still very restricted. One of the key challenges is the overparametrized nature of modern models, enabling complete overfitting of the data even if the labels are randomized, i.e. networks can completely *memorize* all given patterns. While such a memorization capacity seems worrisome, in this work we show that under training protocols that include *data augmentation*, neural networks learn to memorize entirely random labels in a benign way, i.e. they learn embeddings that lead to highly non-trivial performance under nearest neighbour probing. We demonstrate that deep models have the surprising ability to separate noise from signal by distributing the task of memorization and feature learning to different layers. As a result, only the very last layers are used for memorization, while preceding layers encode performant features which remain largely unaffected by the label noise. We explore the intricate role of the augmentations used for training and identify a memorization-generalization trade-off in terms of their diversity, marking a clear distinction to all previous works. Finally, we give a first explanation for the emergence of benign memorization by showing that *malign* memorization under data augmentation is infeasible due to the insufficient capacity of the model for the increased sample size. As a consequence, the network is forced to leverage the correlated nature of the augmentations and as a result learns meaningful features. To complete the picture, a better theory of feature learning in deep neural networks is required to fully understand the origins of this phenomenon.

## 1 INTRODUCTION

Deep learning has made tremendous advances in the past decade, leading to state-of-the-art performance on various learning tasks such as computer vision (He et al., 2016), natural language processing (Devlin et al., 2019) and graph learning (Kipf & Welling, 2017). While some progress has been made regarding the theoretical understanding of these deep models (Arora et al., 2018; Bartlett et al., 2019; 2017; Neyshabur et al., 2015; 2018; Dziugaite & Roy, 2017), the considered settings are unfortunately often very restrictive and the insights made are only qualitative or very loose. One of the key technical hurdles hindering progress is the highly overparametrized nature of neural networks employed in practice, which is in stark contrast with classical learning theory, according to which *simpler hypotheses compatible with the data* should be preferred. The challenge of overparametrization is beautifully illustrated in the seminal paper of Zhang et al. (2017), showing that deep networks are able to fit arbitrary labelings of the data, i.e. they can completely *memorize* all the patterns. This observation renders tools from classical learning theory such as VC-dimension or Rademacher complexity vacuous and new avenues to investigate this phenomenon are needed. The random label experiment has been applied as a sanity check for new techniques (Arora et al., 2018; 2019a; Bartlett et al., 2017; Dziugaite & Roy, 2017), where an approach is evaluated based on its ability to distinguish between networks that memorize or truly learn the data. From a classical perspective, memorization is thus considered as a bug, not a feature, and goes hand in hand with bad generalization.

In this work we challenge this view by revisiting the randomization experiment of Zhang et al. (2017) with a slight twist: we change the training protocol by adding *data augmentation*, a standard practice used in almost all modern deep learning pipelines. We show that in this more practical

setting, the story is more intricate;

*Neural networks trained on random labels with data augmentation learn useful features!*

More precisely, we show that probing the embedding space with the nearest neighbour algorithm of such a randomly trained network admits highly non-trivial performance on a variety of standard benchmark datasets. Moreover, such networks have the surprising ability to separate signal from noise, as all layers except for the last ones focus on feature learning while not fitting the random labels at all. On the other hand, the network uses its last layers to learn the random labeling, at the cost of clean accuracy, which strongly deteriorates. This is further evidence of a strong, implicit bias present in modern models, allowing them to learn performant features even in the setting of complete noise. Inspired by the line of works on benign overfitting (Bartlett et al., 2020; Sanyal et al., 2021; Frei et al., 2022), we coin this phenomenon *benign memorization*. We study our findings through the lens of capacity and show that under data augmentation, modern networks are forced to leverage the correlations present in the data to achieve memorization. As a consequence of the label-preserving augmentations, the model learns invariant features which have been identified to have strong discriminatory power in the field of self-supervised learning (Caron et al., 2021; Grill et al., 2020; Bardes et al., 2022; Zbontar et al.; Chen & He, 2021).Specifically, we make the following contributions:

- We make the surprising observation that learning under complete label noise still leads to highly useful features (*benign memorization*), showing that memorization and generalization are not necessarily at odds.
- We show that deep neural networks exhibit an astonishing capability to separate noise and signal between different layers, fitting the random labels only at the very last layers.
- We highlight the intricate role of augmentations and their interplay with the capacity of the model class, forcing the network to learn the correlation structure.
- We interpret our findings in terms of invariance learning, an objective that has instigated large successes in the field of self-supervised learning.

## 2 RELATED WORK

**Memorization.** Our work builds upon the seminal paper of Zhang et al. (2017) which showed how neural networks can easily memorize completely random labels. This observation has inspired a multitude of follow-up works and the introduced randomization test has become a standard tool to assess the validity of generalization bounds (Arora et al., 2018; 2019a; Bartlett et al., 2017; Dziugaite & Roy, 2017). The intriguing capability of neural networks to simply memorize data has inspired researchers to further dissect the phenomenon, especially in the setting where only a subset of the targets is randomized. Arpit et al. (2017) studies how neural networks tend to learn shared patterns first, before resorting to memorization when given real data, as opposed to random labels where examples are fitted independently. Feldman & Zhang (2020) study the setting when real but "long-tailed" data is used and show how memorization in this case can be beneficial to performance. Maennel et al. (2020); Pondenkandath et al. (2018) on the other hand show how pre-training networks on random labels can sometimes lead to faster, subsequent training on the clean data or novel tasks. Finally, Zhang et al. (2021) show how training on random labels can be valuable for neural architecture search. In all these previous works, data augmentation is excluded from the training pipeline. For partial label noise, it is well-known in the literature that neural networks exhibit surprising robustness (Rolnick et al., 2017; Song et al., 2020; Patrini et al., 2017) and generalization is possible. We highlight however that this setting is distinct from *complete* label noise, which we study in this work. Finally, Dosovitskiy et al. (2014) study the case where each sample has a unique label and achieve strong performance under data augmentation. This setting is again very different from ours as two examples never share the same label, making the task significantly simpler and distinct from memorization.

**Data augmentation.** Being a prominent component of deep learning applications, the benefits of data augmentation have been investigated theoretically in the setting of clean labels (Chen et al., 2020b; Dao et al., 2019; Wu et al., 2020; Hanin & Sun, 2021). The benefits of data augmentation have been verified empirically when only a subset of the data is corrupted (Nishi et al., 2021). On the other hand, investigations with pure label noise where no signal remains in the dataset are absent in

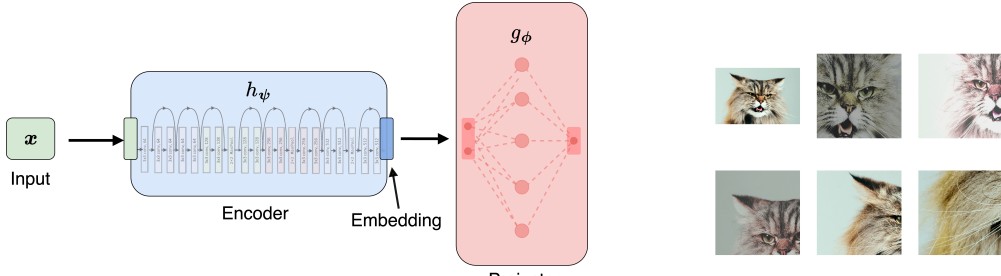

(a) Visualization of an encoder-projector pair. The encoder is typically chosen as a *ResNet18*, while the projector is a one hidden-layer MLP.

(b) Illustration of standard data augmentation. Notice how each augmentation indeed preserves the label "cat".

Figure 1: (Left) Encoder-projector pair. (Right) Standard data augmentations.

the literature. Finally, we want to highlight the pivotal role of data augmentation in self-supervised learning frameworks (Caron et al., 2021; Grill et al., 2020; Bardes et al., 2022; Zbontar et al.; Chen & He, 2021; HaoChen et al., 2021), where it facilitates learning of invariant features.

## 3  BACKGROUND

**Setting.** In this work we consider the standard classification setting, where we are given $n \in \mathbb{N}$ i.i.d. samples $(\boldsymbol{x}_1, \boldsymbol{y}_1), \ldots, (\boldsymbol{x}_n, \boldsymbol{y}_n) \overset{i.i.d.}{\sim} \mathcal{D}$ from some data distribution $\mathcal{D}$, consisting of inputs $\boldsymbol{x} \in \mathbb{R}^d$ and one-hot targets $\boldsymbol{y} \in \mathbb{R}^C$, each encoding one out of $C$ classes. We consider a family of parametrized functions (in this work, neural networks) $f_{\boldsymbol{\theta}} : \mathbb{R}^d \to \mathbb{R}^C$, where $\boldsymbol{\theta} \in \Theta$ denotes the (concatenated) weights from some space $\Theta \subseteq \mathbb{R}^p$ where $p$ is the total number of parameters. Moreover, we specify a loss function $l : \mathbb{R}^C \times \mathbb{R}^C \to \mathbb{R}_+$ which measures the discrepancy between a prediction $f_{\boldsymbol{\theta}}(\boldsymbol{x})$ and its corresponding target $\boldsymbol{y}$, i.e. $l(f_{\boldsymbol{\theta}}(\boldsymbol{x}), \boldsymbol{y})$. We then perform learning by minimizing the empirical loss $\hat{L}$ as a function of the parameters $\boldsymbol{\theta}$,

$$\hat{\boldsymbol{\theta}} := \mathrm{argmin}_{\boldsymbol{\theta} \in \Theta} \hat{L}(\boldsymbol{\theta}) := \mathrm{argmin}_{\boldsymbol{\theta} \in \Theta} \sum_{i=1}^{n} l(f_{\boldsymbol{\theta}}(\boldsymbol{x}_i), \boldsymbol{y}_i) \qquad (1)$$

using some form of stochastic gradient descent and measure the resulting generalization error $L(\hat{\boldsymbol{\theta}}) = \mathbb{E}_{(\boldsymbol{x}, \boldsymbol{y}) \sim \mathcal{D}} \left[ l(f_{\hat{\boldsymbol{\theta}}}(\boldsymbol{x}), \boldsymbol{y}) \right]$. We denote by $p_{\boldsymbol{X}}$ the marginal density of the inputs and by $p_{\boldsymbol{Y}|\boldsymbol{X}}$ the conditional density of the labels given an input. $p_{\boldsymbol{Y}|\boldsymbol{X}}$ encodes the statistical relationship between an input $\boldsymbol{x}$ and the associated label $\boldsymbol{y}$. As typical in practice, we assume that we are in the so-called interpolation regime (Ma et al., 2018), i.e. we assume that stochastic gradient descent can find a parameter configuration $\hat{\boldsymbol{\theta}}$ that achieves zero training loss (see Eq. 1).

**Architecture.** Throughout this work, we consider networks composed of an encoder $h_{\boldsymbol{\psi}} : \mathbb{R}^d \to \mathbb{R}^m$ and a projector $g_{\boldsymbol{\phi}} : \mathbb{R}^m \to \mathbb{R}^C$, where both $\boldsymbol{\psi}$ and $\boldsymbol{\phi}$ denote the parameters of the respective building block. As an encoder, we typically employ modern convolutional architectures such as *ResNets* (He et al., 2016) or *VGG* (Simonyan & Zisserman, 2014), excluding the final fully-connected layers, while the projector is usually an MLP with one hidden layer. We illustrate such a network in Fig. 1a. Such architectures have become very popular in the domain of feature learning and are extensively used in unsupervised and self-supervised learning (Caron et al., 2021; Grill et al., 2020; Bardes et al., 2022; Zbontar et al.). In this work, we are interested in assessing the quality of the encoder's features $h_{\boldsymbol{\psi}}$ when the network $f_{\boldsymbol{\theta}} = g_{\boldsymbol{\phi}} \circ h_{\boldsymbol{\psi}}$ is trained on random labels.

**Probing.** To evaluate the embeddings $h_{\boldsymbol{\psi}}$, we apply nearest-neighbour-based and linear *probing*, which refers to performing $K$-nearest-neighbour (or linear) classification based on the embeddings $\{(h_{\boldsymbol{\psi}}(\boldsymbol{x}_i), \boldsymbol{y}_i)\}_{i=1}^{n}$. We fix the number of neighbours to $K = 20$ throughout this work unless otherwise specified. Probing measures how useful a given set of features $h_{\boldsymbol{\psi}}$ is for a fixed task. A special case we will often consider in this work is probing the encoder of a network at initialization, which we refer to as probing at initialization. Due to the lack of feature learning in probing, the resulting performance is very dependent on the quality of the input representations $h_{\boldsymbol{\psi}}$. Such method

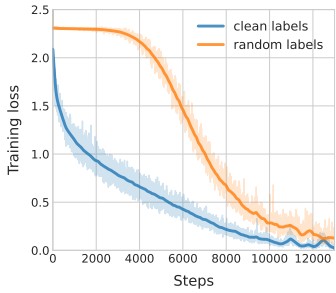

| Layer | Initialization | End of training |
|---|---|---|
| Layer 1 | 36.9 | 34.3 |
| Layer 2 | 36.4 | 33.5 |
| Layer 3 | 36.5 | 30.4 |
| Layer 4 | 36.2 | 14.9 |
| Embedding | 40.2 | 12.4 |
| Projector hidden layer | 38.4 | 10.5 |
| Output | 24.2 | 10.5 |

Figure 2: Fitting random labels on unaugmented data on *CIFAR10* with a *ResNet18* is not significantly slower than fitting clean labels.

Table 1: $K$-NN probing accuracies (percentage) for a *ResNet18* at initialization versus at the end of training on random labels without augmentations on *CIFAR10*. The four layers correspond to the intermediate representations after each stage of a *ResNet*.

has been used to assess the quality of a given embedding space in various works, for instance Alain & Bengio (2017); Chen et al. (2020a); Caron et al. (2021); Grill et al. (2020); Bardes et al. (2022); Zbontar et al..

**Random labels and memorization.** Denote by $e_l \in \mathbb{R}^C$ the $l$-th unit vector, i.e. the one-hot encoding of class $l$. Zhang et al. (2017) introduced a randomization test, where any clean label $y_i \in \{e_1, \ldots, e_C\}$ in the training set is replaced by a random one-hot encoding, $\tilde{y}_i = e_l$ where $l \sim \mathcal{U}(\{1, \ldots, C\})$ with $\mathcal{U}$ denoting the uniform distribution over the discrete set $\{1, \ldots, C\}$. Throughout this text, we will denote randomized variables with $\sim$ on top. Notice that such an intervention destroys any statistical relationship between inputs and targets, i.e. $p_{Y|X} = p_Y$. As a consequence, training on such a dataset should become very difficult as there is no common pattern left to exploit and an algorithm has to resort to a pure memorization approach. Zhang et al. (2017) showed that deep neural networks have an astonishing capacity to perform such memorization tasks, easily overfitting any random label assignment even on large-scale datasets such as *ImageNet* (Deng et al., 2009) without a large increase in training time (see Fig. 2). Even explicit regularizers such as weight decay and *Dropout* (Srivastava et al., 2014) can be used in a standard manner, data augmentation however is excluded from the pipeline. We have reproduced a subset of the results of Zhang et al. (2017) in Table 2, first column. As expected, deep neural networks indeed manage to achieve zero training error across a variety of datasets while not generalizing at all, both with respect to a random test labeling as well as the original, clean test labels. In order to further study the amount of distortion in the resulting embedding space, we apply nearest-neighbour probing with respect to the clean data. More precisely, given the features $h_\psi$ learnt from training on the random label task $\{(x_i, \tilde{y}_i)\}_{i=1}^n$, we apply probing based on the clean training data $\{(x_i, y_i)\}_{i=1}^n$ and evaluate with respect to clean test data. We display the results in Table 1. In line with previous works, (Cohen et al., 2018; Maennel et al., 2020), we find that while very early layers might retain their performance at initialization, a significant drop occurs with increasing depth, further highlighting the lack of feature learning and the *malignant* nature of memorization in this setting. This is in line with observations that early layers learn useful representations (Zhang et al., 2019).

**Data Augmentation.** A standard technique present in almost all computer vision pipelines is data augmentation. We consider transformations $T : \mathbb{R}^d \to \mathbb{R}^d$, that take a given input $x$ and produce a new augmentation $\bar{x} := T(x)$, which by design, should preserve the associated label, i.e. $\bar{y} = y$. Such transformations are usually given as a composition of smaller augmentations including random crops, flips, color-jittering etc. In Fig. 1b we show a set of different augmentations of the same underlying image $x$. Notice how these transformations indeed leave the associated label invariant. We denote by $\mathcal{T}$ the set of all possible augmentations. Data augmentation is usually applied in an online fashion, i.e. at every step of gradient descent, we uniformly sample a fresh transformation $T \sim \mathcal{U}(\mathcal{T})$ and propagate it through the network. We highlight that data augmentation is a standard technique necessary for state-of-the-art performance for a variety of vision tasks. Indeed, the top five leaders on *ImageNet*[1] (Yu et al., 2022; Dai et al., 2021; Zhai et al., 2021; Pham et al., 2021; Liu et al., 2021) all rely on some form of data augmentation in their training pipeline. If one hence

---

[1]https://paperswithcode.com/sota/image-classification-on-imagenet

| DATASET | MODEL | RANDOM | RANDOM + DA | CLEAN | CLEAN + DA | INIT |
|---------|-------|--------|-------------|-------|------------|------|
| *CIFAR10* | *ResNet18* | 12.4 | 76.2 | 83.6 | 91.3 | 40.6 |
| | *VGG11* | 15.0 | 71.9 | 83.0 | 89.0 | 44.3 |
| *CIFAR100* | *ResNet18* | 5.0 | 42.7 | 50.5 | 68.2 | 16.6 |
| | *VGG11* | 6.0 | 46.4 | 51.2 | 59.0 | 18.9 |
| *TinyImageNet* | *ResNet18* | 2.4 | 33.4 | 38.3 | 46.6 | 4.6 |
| | *VGG11* | 1.2 | 29.4 | 34.6 | 43.8 | 5.4 |

Table 2: $K$-NN probing accuracies (in percentage) of the embeddings for various datasets under different settings. DA refers to training under standard data augmentation. INIT refers to the performance at initialization. Except for INIT, all models reach perfect (unaugmented) training accuracy.

wants to study the memorization potential of neural networks in practical settings, data augmentation needs to be considered. We notice that the results of Zhang et al. (2017) and subsequent studies on the properties of memorization under random labels (Arpit et al., 2017; Maennel et al., 2020) were obtained without the use of data augmentation, leaving the memorization picture thus incomplete.

## 4 BENIGN MEMORIZATION

In this section, we present the curious phenomenon of benign memorization i.e. how neural networks manage to completely fit random labels under data augmentation, while at the same time learning predictive features for downstream tasks. Let us first formally introduce the terms *benign* and *malign* memorization, which are central to the results of this work. In the following, $\mathcal{S} := \{(\boldsymbol{x}_i, \boldsymbol{y}_i)\}_{i=1}^n$ denotes the original *clean* dataset and $\tilde{\mathcal{S}} = \{(\boldsymbol{x}_i, \tilde{\boldsymbol{y}}_i)\}_{i=1}^n$ its randomly labeled version.

**Definition 4.1** *We call an encoder-projector pair $(h_{\phi_*}, g_{\psi_*})$ a **memorization** of $\tilde{\mathcal{S}}$, if $f_*$ perfectly fits $\tilde{\mathcal{S}}$. Moreover, we call $(h_{\phi_*}, g_{\psi_*})$ a **malign** memorization if additionally, probing of $h_{\phi_*}$ on $\mathcal{S}$ does not improve over probing at initialization. On the contrary, we call $(h_{\phi_*}, g_{\psi_*})$ a **benign** memorization of $\tilde{\mathcal{S}}$ if probing of $h_{\phi_*}$ on $\mathcal{S}$ outperforms probing at initialization.*

As highlighted in Sec. 3 and shown in Table 2, memorizing solutions found with stochastic gradient descent without data augmentation on standard vision benchmarks are of malign nature. This is very intuitive, as randomizing the targets destroys any signal present in the dataset and thus generalization seems impossible. We show now how including data augmentation in the training pipeline completely reverses this observation.

**Training details.** In all subsequent experiments involving data augmentations, we use the standard transformations employed in self-supervised learning frameworks such as Chen et al. (2020c); Grill et al. (2020); Chen & He (2021). These consist of a composition of random crops, color-jittering, random greyscaling, and random horizontal flips, leading to a diverse set of transformations. Moreover, we rely on mixup augmentations Zhang et al. (2018), where two images $\boldsymbol{x}_1$, $\boldsymbol{x}_2$ are combined into a linear interpolation, according to some weighting $\alpha \in [0, 1]$, i.e. $\bar{\boldsymbol{x}} := \alpha \boldsymbol{x}_1 + (1 - \alpha)\boldsymbol{x}_2$. The corresponding label $\bar{\boldsymbol{y}}$ is accordingly subject to the same linear combination, i.e. $\bar{\boldsymbol{y}} = \alpha \boldsymbol{y}_1 + (1 - \alpha)\boldsymbol{y}_2$ where labels are represented as their one-hot encodings. We use the standard vision datasets *CIFAR10* and *CIFAR100* (Krizhevsky & Hinton, 2009), as well as *TinyImageNet* (Le & Yang, 2015). For more details, we refer the reader to Appendix F.

**Benign Memorization.** We display the results of training under data augmentation in Table 2. We observe that, surprisingly, nearest-neighbour probing of the learnt embeddings leads to clearly non-trivial performance, strongly improving over the models trained under random labels without data augmentation. Moreover, we strongly outperform probing at initialization, showing that indeed rich feature learning is happening. As a consequence, data augmentation does not simply prevent malign memorization by preserving the signal at initialization but actually leads to learning from the data. On the other hand, it holds that the projector achieves perfect training accuracy on the random labels and the network thus indeed memorizes the training data perfectly. Under data augmentation, deep models hence exhibit benign memorization. In Appendix C, Fig. 9, we further underline the

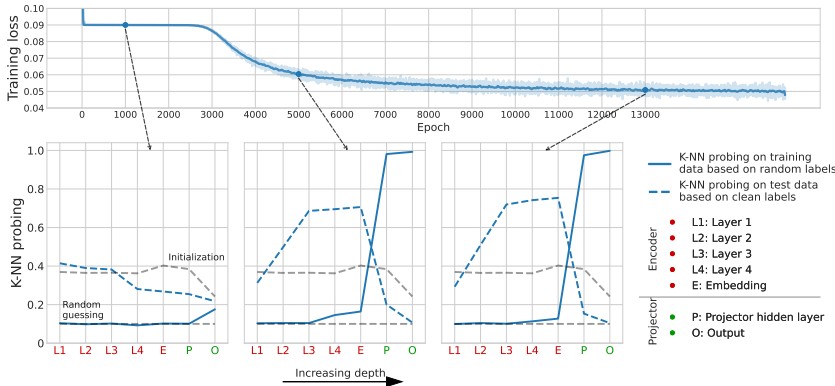

Figure 3: $K$-NN probing of a *ResNet18* on *CIFAR10*, based on (flattened) representation of different layers. Dashed blue lines indicate $K$-NN accuracies fitted on the clean training data and evaluated w.r.t. clean test data. Solid lines indicate $K$-NN training accuracies based on the random labels. Notice the sharp phase transition when reaching the projector layers"P" and "O", i.e. the projector reaches strong performance on the noisy labels (dashed) but randomly guesses on the clean data (solid). The opposite happens for the embedding layer, reaching $76\%$ $K$-NN accuracy.

utility of the learnt features by evaluating their performance under transfer learning. We remark that this is, to the best of our knowledge, the first work showing that learning under random labels can lead to strong downstream performance, highlighting that memorization and generalization are not necessarily at odds. We notice that the training time under randomized targets together with data augmentation increases significantly, both compared to training under clean labels as well as fitting random labels without augmentations. In Fig. 3 we show the evolution of both the training loss as well as nearest-neighbour probing test accuracy as a function of the number of epochs. Observe that for as long as 3000 epochs, neither training loss nor probing accuracy show any progress but then suddenly start to rapidly improve.

We stress that our goal is not to compete with standard training under clean labels, it can be seen in Table 2 that there is a large gap between the two methods. Instead we rather aim for a deeper understanding of how deep networks generalize. We show that generalization, contrary to prior beliefs, remains possible even in the most adversarial setting of complete label noise.

**Signal-Noise Separation.** We now further inspect models trained under random labels and data augmentation, with an emphasis on how the noise stemming from the random labeling affects different parts of the network. To gain insights into this, we perform nearest-neighbour probing of different layers in the network, with the twist that we fit the $K$-NN classifier both with respect to the clean training labels, as well as with respect to the random labels. This way we can assess how much a given feature has adapted to the particular structure (clean vs. random). We visualize the results of such a clean and noisy probing strategy in Fig. 3 at different stages of training. Surprisingly, we can see a striking separation between feature learning and memorization within the architecture; noisy probing only starts to improve over random guessing once we reach the projector, whereas the encoder remains largely unaffected. On the other hand, clean probing outperforms probing at initialization throughout the entire embedding stage but sharply decays once we get to the projector. Some previous work highlighted that even when training on random labels without data augmentation, the very first layers learn data dependent features which lead to subsequent fast re-training on clean data Maennel et al. (2020). We give an interpretation of this finding in Sec. 5.2. We further investigate the role of the projector in Appendix C, Fig. 7, finding that higher widths lead to better performance in general.

## 5 THE ROLE OF AUGMENTATIONS

The empirical evidence in the previous section highlights the crucial role of data augmentation for benign memorization. In this section, we aim to gain insights into the phenomenon by investigating more closely what properties of data augmentation are essential for benign memorization and how "ideal" augmentations might strongly differ when learning with clean or random labels.

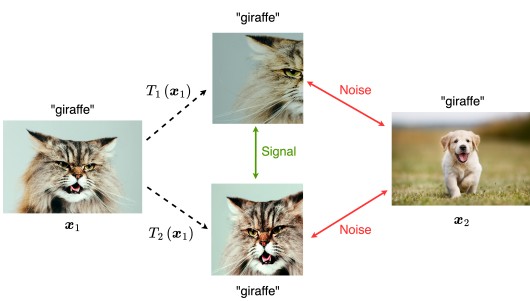

| Dataset Size | 1000 | 50000 |
|---|---|---|
| Random | 11.41 | 12.41 |
| Random + i.i.d. DA | 12.41 | - |
| Clean | 36.75 | 83.66 |
| Clean + i.i.d. DA | 82.73 | - |
| Clean + DA | 69.7 | 91.3 |
| Random + DA | 54.7 | 76.2 |

Figure 4: Illustration of the thought experiment in Sec. 5. The true cat $x_1$ labeled as "giraffe" is augmented, leading to a positive signal between the augmentations but to more noise due to $x_2$, that has as true label "dog".

Table 3: $K$-NN probing accuracies of the embeddings of a *ResNet18* for *CIFAR10* with and without i.i.d. augmentations under clean and random labels. 50000 refers to full *CIFAR10* for reference.

**Label Preservation.** The first important characteristic of augmentations is their label-preserving nature, i.e. by augmenting an image $(x, y)$ twice to produce $(T_1(x), y), (T_2(x), y)$, we have effectively added information, as indeed $T_1(x)$ and $T_2(x)$ share the same label. Unfortunately, this reasoning is flawed in the setting of random labels, or at the least only forms part of a larger picture, as we show in the following *thought experiment*. Consider two "original" samples $(x_1, \tilde{y})$ and $(x_2, \tilde{y})$ which happen to share the same random label $\tilde{y}$ but in truth have distinct labels $y_1 \neq y_2$. In this case, forming augmentations $(T_1(x_1), \tilde{y})$ and $(T_2(x_1), \tilde{y})$ might lead to some correct signal as $T_1(x_1)$ and $T_2(x_1)$ share the same, true label, but at the same time leads to more distortion as $x_2$ has the same, random label, reinforcing the wrong correlation even more. As a consequence, separating noise from signal remains equally challenging as before. We illustrate the argument in Fig. 4. To check this hypothesis, we consider the extreme case where augmentations $T \in \mathcal{T}$ produce a new, i.i.d. sample $T(x)$ that shares the same, true label with $x$. In a sense, this is the ideal augmentation and leads to the highest information gain. We implement such i.i.d. augmentations by only using a subset of the training data while using the remainder to assign to each training point $B$ potential, independent examples with the same, true label. We choose the subset size as $s = 1000$ and the number of augmentations as $B = 50$ for *CIFAR10* and train the same models as in Sec. 4, both under random and clean labels. We display the results in Table 4. Notice that for clean label training, such augmentations increase the training set size from 1000 to $1000 \times 50 = 50000$, thus leading to almost identical performance as training on the full dataset. Counter-intuitively, but for the reasons outlined above, training under random labels severely suffers under such ideal, independent augmentations and leads to malign memorization. In fact, this experiment is equivalent to fitting random labels without augmentations on the full training set where malign memorization occurs, as seen in Sec. 4.

This thought experiment demonstrates that under random labels, augmentations seem to play a very distinct role compared to the clean setting and label preservation in itself is not enough to guarantee benign memorization. This raises the following question:

*What other properties of augmentations, besides label preservation, cause benign memorization?*

We hypothesize that the origins of the phenomenon lie at the interplay between the highly correlated nature of augmentations and the inflated effective sample size that exceeds the model capacity (Sec 5.1), forcing the model to learn meaningful features (Sec 5.2).

## 5.1 Going Beyond the Model Capacity

We now study the phenomenon of benign memorization from the view point of capacity of a model class. Intuitively speaking, the capacity $\mathcal{C}_t$ of a model captures how many distinct datapoints we can fit in $t$ gradient steps, even if the corresponding targets are completely randomized. We refer to Appendix D.1 for the formal definition adopted here. If a model has enough capacity, it can potentially memorize all the patterns in the dataset. As seen in Zhang et al. (2017), deep models used in practice in conjunction with standard datasets, operate below the capacity threshold but nevertheless, they do not "abuse" their power and fit clean data in a meaningful way. On the other

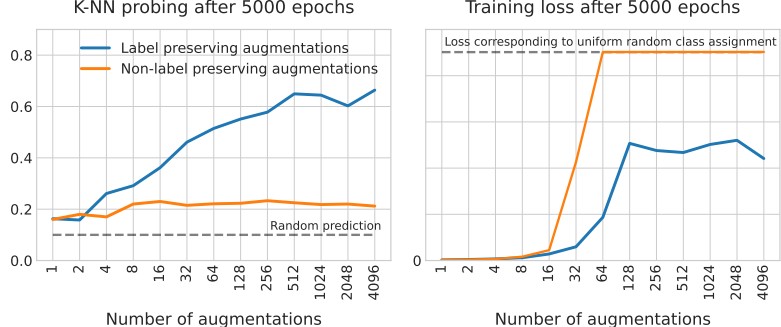

Figure 5: Illustration of nearest-neighbour probing accuracy (left) and (running) training loss (right) of a *ResNet18* on *CIFAR10* with increasing number of standard augmentations. We consider both label-preserving (blue) and completely random labeling of augmentations (red).

hand, by using data augmentation we inflate the number of samples and consequently operate above the capacity level. As a result, a model needs to *efficiently* encode augmentations (pure memorization is not possible) and hence meaningful features need to be learnt.

As seen in Sec. 3, standard datasets such as *CIFAR10* have size $n$ below the capacity $\mathcal{C}_t$ of modern networks such as *ResNets*. When using data augmentation however, we now show that the resulting dataset exceeds the capacity.

**Inflated Sample Size.** Consider a set of augmentations $\mathcal{T}$ and assume that it has a finite size, $B := |\mathcal{T}| < \infty$. Augmenting thus leads to a larger effective dataset $\mathcal{S}_{\text{aug}}$ of size $Bn$. We can now study whether the capacity of standard networks trained by GD for a fixed number of epochs exceeds such an augmented dataset by varying $B$ and randomly labeling each sample. We pre-compute a fixed set of $B$ random augmentations for each sample for varying $B$ and attempt to memorize a random labeling, where labels of augmentations are **not** preserved. We use the same setup as in Sec. 4 and augment *CIFAR10* using the standard augmentations in the way described in Sec. 4. We display the results in Fig. 5. We see that indeed, overfitting the random labels becomes more difficult as we increase $B$ and actually infeasible for a fixed number of gradient steps $t = 5000 \times 196$ (fixed batch size on *CIFAR10*). Moreover, notice that in the standard setting of online augmentations, this observation becomes even more drastic as $B$ increases over time if typical, continuous augmentations such as color jittering are included in the pipeline. We hypothesize that malign memorization under such an augmentation strategy thus becomes infeasible.

**Learn if you must.** We now investigate the influence of capacity on the resulting probing accuracy, in the case where augmentations are label-preserving and hence offer valuable signal. We thus consider the same setup as in the previous paragraph for varying number of augmentations $B$ while assigning the same label to augmentations of the same image. We show the results in Fig. 5 on the left. We observe that as the number of augmentations $B$ increases, probing accuracy improves and eventually surpasses probing at initialization. We hypothesize that as we approach and eventually surpass capacity, the model is increasingly forced to leverage the signal in augmentations and thus learns more and more relevant features. As we increase $B$, we saturate the information gain provided by the augmentations, the signal becomes redundant and performance starts to plateau.

## 5.2 What can you Learn from Augmentations?

While we have seen that the model is forced to leverage the signal in augmentations, it remains unclear why this leads to high-quality embeddings. We now show how augmentations encourage features to become invariant, a property that has been identified in SSL to be strongly predictive.

**Normalized Invariance.** To measure the invariance of a function $q : \mathbb{R}^d \to \mathbb{R}^a$, we introduce the following quantity:

$$I(\boldsymbol{x}; q, \mathcal{T}) := \frac{\mathbb{E}_{T_1, T_2 \sim \mathcal{U}(\mathcal{T})} \|q(T_1(\boldsymbol{x})) - q(T_2(\boldsymbol{x}))\|_2}{\mathbb{E}_{\boldsymbol{x}' \neq \boldsymbol{x}} \|q(\boldsymbol{x}) - q(\boldsymbol{x}')\|_2}. \tag{2}$$

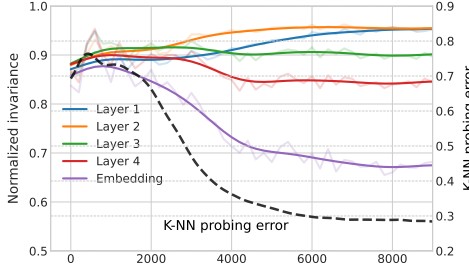

| Method | Normalized invariance |
|---|---|
| Initialization | 0.867 |
| Clean | 0.912 |
| Clean + DA | 0.691 |
| Random | 0.989 |
| Random + DA | 0.680 |

Figure 6: Averaged normalized invariance of a *ResNet18* on *CIFAR10*, as a function of epochs, lower value means more invariant. The dashed line represents $K$-NN probing error of the embedding layer.

Intuitively, $I(\boldsymbol{x}; q, \mathcal{T})$ captures the features' similarity of augmentations of the same datapoint $\boldsymbol{x}$ compared to the representations of different datapoints $\boldsymbol{x}' \neq \boldsymbol{x}$. Model's invariance implies $I(\boldsymbol{x}; q, \mathcal{T}) = 0$, hence different augmentations are mapped to the same datapoint, while the model is still able to meaningfully distinguish between the representations of different datapoints (i.e. $\mathbb{E}_{\boldsymbol{x}' \neq \boldsymbol{x}} \|q(\boldsymbol{x}) - q(\boldsymbol{x}')\| \neq 0$). In Fig. 6, we show how $I(\boldsymbol{x}; q, \mathcal{T})$ for different layers correlates with probing performance and indeed decreases over time. Due to its better implicit bias (*ResNet* vs MLP), invariance is largely learnt in the encoder, leading to the striking signal-noise separation identified in Sec. 4. These results suggest that when the model has insufficient capacity to memorize the (augmented) samples, it learns to be more invariant with respect to the label-preserving augmentations. Consequently, this mechanism reduces the "effective sample size" and allows the model to fit the data in a *benign* (i.e. augmentation-invariant) way. But why do invariant features imply a better clustering in embedding space as evidenced by a high $K$-NN accuracy? This is a heavily researched topic with several plausible theories in the area of self-supervised learning (SSL) (Saunshi et al., 2022; Arora et al., 2019b; Wen & Li, 2021). In Appendix B, we derive a more formal connection between SSL loss and training with random labels. Others have looked at the improved sample complexity caused by incorporating invariances into the model (Bietti et al., 2021; Xiao & Pennington, 2022).

## 6  DISCUSSION AND CONCLUSION

In this work we have identified the surprising phenomenon of benign memorization, demonstrating that generalization and memorization are not necessarily at odds. We put forward an interpretation through the lens of model capacity, where the inflation in sample size forces the model to exploit the correlation structure by learning the invariance with respect to the augmentations. Furthermore, we have shown that invariance learning happens largely in the encoder, while the projector performs the noisy memorization task.

Our findings underline the complicated and mysterious inner workings of neural networks, showing that generalization can be found where we least expect it. To describe benign memorization, a complete generalization theory needs to capture the strong implicit bias built into deep models, which enables a clean separation of noise and signal. Secondly, such a theory needs to incorporate feature learning, as benign memorization only emerges in the encoder. Both those goals remain very challenging. In particular, we speculate that the line of work based on the *neural tangent kernel* (Jacot et al., 2018) which connects SGD-optimized neural networks in the infinite width regime and the realm of kernels cannot explain benign memorization. In fact, in this regime the weights do not move from initialization, thus preventing feature learning. Some recent works have pushed further and study neural networks outside of the kernel regime (Allen-Zhu & Li, 2019; 2020) and we believe that the developed tools could be very helpful in understanding benign memorization. Another promising direction are perturbative finite width corrections to NTKs (Hanin & Nica, 2019; Zavatone-Veth et al., 2021) which also incorporate feature learning. We leave exploring benign memorization in such a mathematical framework as future work.

# 7 REPRODUCIBILITY STATEMENT

We have taken multiple steps to ensure reproducibility of the experiments. We refer the reader to Appendix F for a complete description of the training protocol. We have also released the code as part of the supplementary material, including scripts on how to reproduce our results.

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

| Dataset | Model | Random | Random + DA | Clean | Clean + DA | Init |
|---------|-------|--------|-------------|-------|-----------|------|
| *CIFAR10* | *ResNet18* | 29.8 | 77.0 | 83.9 | 92.5 | 43.8 |
| | *VGG11* | 23.9 | 71.0 | 81.8 | 89.8 | 40.7 |
| *CIFAR100* | *ResNet18* | 10.3 | 44.2 | 52.3 | 69.6 | 18.5 |
| | *VGG11* | 12.7 | 47.5 | 51.5 | 61.4 | 16.9 |
| *TinyImageNet* | *ResNet18* | 4.7 | 33.9 | 39.5 | 47.7 | 3.7 |
| | *VGG11* | 3.7 | 22.3 | 33.0 | 41.3 | 4.4 |

Table 4: Linear probing accuracies (in percentage) of the embeddings for various datasets under different settings. DA refers to training under data augmentation. INIT refers to the performance at initialization.

## A  Linear Probing Results

We replicate the results of Table 4 but with linear probing instead of $K$-NN probing of the embeddings, for the same models and the same datasets. In this case we train a linear classifier on top of the embeddings of the true (unaugmented) training data and evaluate on the left-out test set. We see that random without data augmentation although above guessing is still below the performance at initialization.

## B  Connection to Self-Supervised Learning

In this section, we investigate the connection between non-contrastive SSL (Hua et al., 2021b; Grill et al., 2020; Chen & He, 2021) and training with random labels on the mean squared error (MSE) loss. We start with the following result:

**Lemma B.1** *Fix $B \in \mathbb{N}$ vectors $\boldsymbol{x}_1, \ldots, \boldsymbol{x}_B \in \mathbb{R}^d$ and $\boldsymbol{a} \in \mathbb{R}^d$. Then it holds that*

$$\frac{1}{B} \sum_{i=1}^{B} \|\boldsymbol{x}_i - \boldsymbol{a}\|^2 = \frac{1}{2B^2} \sum_{i,j=1}^{B} \|\boldsymbol{x}_i - \boldsymbol{x}_j\|^2 + \|\boldsymbol{a} - \frac{1}{B} \sum_{i=1}^{B} \boldsymbol{x}_i\|^2$$

**Proof:** We simply expand the first term on the right-hand-side:

$$\frac{1}{2B^2} \sum_{i,j=1}^{B} \|\boldsymbol{x}_i - \boldsymbol{x}_j\|^2 = \frac{1}{2B^2} \sum_{i,j=1}^{B} \|\boldsymbol{x}_i - \boldsymbol{a} + \boldsymbol{a} - \boldsymbol{x}_j\|^2$$

$$= \frac{1}{2B^2} \left( \sum_{i,j=1}^{B} \|\boldsymbol{x}_i - \boldsymbol{a}\|^2 + \|\boldsymbol{x}_j - \boldsymbol{a}\|^2 - 2(\boldsymbol{x}_i - \boldsymbol{a})^T(\boldsymbol{x}_j - \boldsymbol{a}) \right)$$

$$= \frac{1}{B} \sum_{i}^{B} \|\boldsymbol{x}_i - \boldsymbol{a}\|^2 - \frac{1}{B^2} \sum_{i,j=1}^{B} (\boldsymbol{x}_i - \boldsymbol{a})^T(\boldsymbol{x}_j - \boldsymbol{a})$$

$$= \frac{1}{B} \sum_{i}^{B} \|\boldsymbol{x}_i - \boldsymbol{a}\|^2 - \|\boldsymbol{a}\|^2 - \frac{1}{B^2} \sum_{i,j=1}^{n} \boldsymbol{x}_i^T \boldsymbol{x}_j + \frac{2}{B} \sum_{i=1}^{n} \boldsymbol{a}^T \boldsymbol{x}_i$$

On the other hand, we have that

$$\|\boldsymbol{a} - \frac{1}{B} \sum_{i=1}^{B} \boldsymbol{x}_i\|^2 = \|\boldsymbol{a}\|^2 + \frac{1}{B^2} \sum_{i,j=1}^{n} \boldsymbol{x}_i^T \boldsymbol{x}_j - \frac{2}{B} \sum_{i=1}^{n} \boldsymbol{a}^T \boldsymbol{x}_i$$

Hence we see that

$$\frac{1}{2B^2} \sum_{i,j=1}^{B} \|\boldsymbol{x}_i - \boldsymbol{x}_j\|^2 = \frac{1}{B} \sum_{i}^{B} \|\boldsymbol{x}_i - \boldsymbol{a}\|^2 - \|\boldsymbol{a} - \frac{1}{B} \sum_{i=1}^{B} \boldsymbol{x}_i\|^2$$

and re-arranging terms concludes the proof.

We now apply this result in the case where the label plays the role of $\boldsymbol{a}$ and $\boldsymbol{x}_i$'s play the role of different augmentations. Let us thus consider $B$ augmentations $T_1, \ldots, T_B$ and $n \in \mathbb{N}$ samples $\boldsymbol{x}_1, \ldots, \boldsymbol{x}_n \in \mathbb{R}^d$. Moreover we have some labels $\boldsymbol{y}_1, \ldots, \boldsymbol{y}_n \in \mathbb{R}^K$ that could be completely random. Using the previous result we can write the (random) supervised loss (assuming mean-squared error) as follows:

**Theorem B.2** *Denote the supervised loss under data augmentation as*

$$\hat{L}^{super}(\boldsymbol{\theta}) = \frac{1}{nB} \sum_{i=1}^{n} \sum_{a=1}^{B} \left\| f_{\boldsymbol{\theta}}(T_a(\boldsymbol{x}_i)) - \boldsymbol{y}_i \right\|^2$$

*Then we can decompose it into the following two terms,*

$$\hat{L}^{super}(\boldsymbol{\theta}) = \underbrace{\frac{1}{2nB^2} \sum_{i=1}^{n} \sum_{a,b=1}^{B} \left\| f_{\boldsymbol{\theta}}(T_a(\boldsymbol{x}_i)) - f_{\boldsymbol{\theta}}(T_b(\boldsymbol{x}_i)) \right\|^2}_{=:\text{Inv}(\boldsymbol{\theta}) \geq 0} + \underbrace{\frac{1}{n} \sum_{i=1}^{n} \left\| \boldsymbol{y}_i - \frac{1}{B} \sum_{a=1}^{B} f_{\boldsymbol{\theta}}(T_a(\boldsymbol{x}_i)) \right\|^2}_{=:\text{Bias}(\boldsymbol{\theta}) \geq 0}$$

Notice that in $\text{Inv}(\boldsymbol{\theta})$, we group augmentations of the same input $\boldsymbol{x}_i$ together, measuring thus how invariant a given model $f_{\boldsymbol{\theta}}$ is w.r.t. to the augmentations. This is the positive signal illustrated in the thought experiment in Fig. 4. Interestingly, minimizing $\text{Inv}(\boldsymbol{\theta})$ is a core ingredient for so-called non-contrastive self-supervised learning methods (Hua et al., 2021b; Grill et al., 2020; Chen & He, 2021).

The bias term on the other hand is influenced by the random labeling. To better understand the bias term, define $N_c := \{i \in \{1, \ldots, n\} : \boldsymbol{y}_i = \boldsymbol{e}_c\}$, i.e. all samples that have label $c$. Furthermore, let $\bar{f}_{\boldsymbol{\theta}}(\boldsymbol{x}_i) = \frac{1}{B} \sum_{a=1}^{B} f_{\boldsymbol{\theta}}(T_a(\boldsymbol{x}_i))$ be the model average over all the $B$ augmentations. We can show that we can express the bias as

$$\text{Bias}(\boldsymbol{\theta}) = \frac{1}{n} \sum_{c=1}^{C} \sum_{i \in N_c} \left\| \boldsymbol{e}_c - \bar{f}_{\boldsymbol{\theta}}(\boldsymbol{x}_i) \right\|^2.$$

Notice that as in a random labeling experiment, the number of classes $C$ can be considered as a hyperparameter, hence if we choose $C >> n$, then each sample $\boldsymbol{x}_1, \ldots, \boldsymbol{x}_n$ is assigned a different class with high probability. In this case, the "bad bias" given by assigning the same label to samples of different classes vanishes, and one has:

$$\text{Bias}(\boldsymbol{\theta}) = \frac{1}{n} \sum_{i=1}^{n} \left\| \boldsymbol{e}_i - \bar{f}_{\boldsymbol{\theta}}(\boldsymbol{x}_i) \right\|^2,$$

where w.l.o.g we have re-ordered the labels such that the $i$-th label is assigned to sample $\boldsymbol{x}_i$. In this limiting case, the bias term represents a contrastive term that encourages the average model $\bar{f}_{\boldsymbol{\theta}}(\boldsymbol{x}_i)$ to be different from the other samples $\boldsymbol{x}_j \neq \boldsymbol{x}_i$ and their augmentations. In this sense, this term can be related to contrastive SSL. On the other hand — unlike contrastive SSL — the distance between the average models of two datapoints cannot be arbitrarily large, and once the loss is driven to zero, $\text{Bias}(\boldsymbol{\theta}) = 0$ and it is trivial to see that $\left\| \bar{f}_{\boldsymbol{\theta}}(\boldsymbol{x}_i) - \bar{f}_{\boldsymbol{\theta}}(\boldsymbol{x}_j) \right\|^2 = 2$.

We refer the reader to Appendix C for experiments with varying number of classes. We observe that, as hypothesized, the influence of the bias diminishes and performance increases as a function of the number of classes $C$.

## C  MORE EXPERIMENTS

In the following we present more empirical evidence that may help better interpret our findings.

**Role of the Projector.** We visualize in Fig. 7 the effect of varying the size of the hidden layer in the projector. There is a threshold under which learning is impossible (or at least very hard given

a fixed computational budget). Surprisingly, using a larger projector seems beneficial beyond the point where we manage to decrease the loss significantly and achieve memorization of the original dataset.

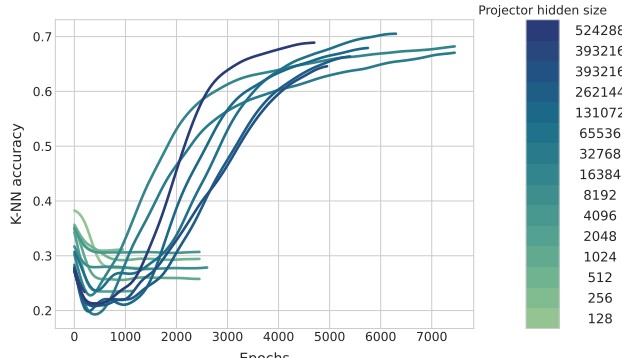

Figure 7: Nearest-neighbour probing accuracy as function of the size of the projector.

**The role of the number of random classes.** So far we assigned to each sample a random label in $\mathbb{R}^C$, where $C$ is the number of classes in the original dataset. In principle, increasing the number of classes decreases the strength of the noise provided to the model. We verify this in Fig. 8, where we vary the number of possible classes that each sample is randomly assigned to. In the extreme case, every sample is assigned its unique class label $\boldsymbol{y}_i = \boldsymbol{e}_i$, where $\boldsymbol{e}_i \in \mathbb{R}^n$, recovering the result of Dosovitskiy et al. (2014) which we refer to as 'Per sample'. We observe that while increasing number of classes performance improves (as less noisy assignments are made). We refer the reader to Appendix B for a more formal connection.

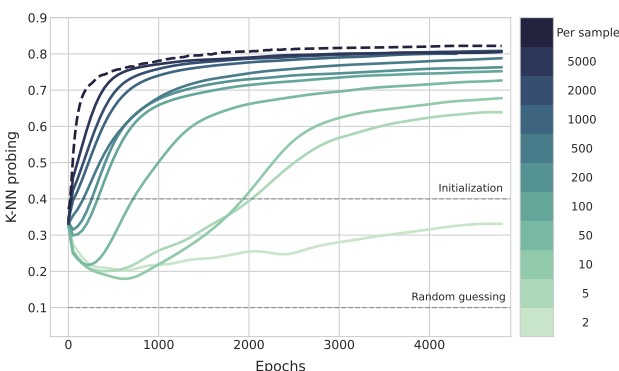

Figure 8: Nearest-neighbour probing accuracy as a function of the number of classes.

**Transferability.** Features learnt with random labels are also useful when evaluated on different datasets. In Fig. 9 on the right-hand-side, we transfer the features learnt on *CIFAR10*, to *CIFAR100* and *STL10* (Coates et al., 2011). We observe that we can achieve strong downstream performance, highlighting the strength of the features learnable under random labels. Moreover, we see that using data augmentation is crucial when label noise is high as without data augmentation

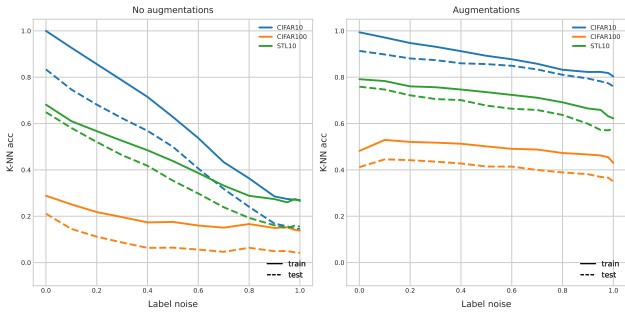

Figure 9: Nearest-neighbour probing accuracy for transfer learning to different datasets as a function of label noise. A *ResNet18* is trained based on random labels on *CIFAR10* with (right) and without (left) data augmentation.

**Dependence on noise and speed of convergence.** Augmentations have been shown to be especially useful under the presence of heavy label noise. In Fig. 10 we visualize the performance achieved during the first stage of training under varying level of label noise. Label noise specifies the percentage of labels that are randomly permuted versus the percentage of labels that are kept the same.

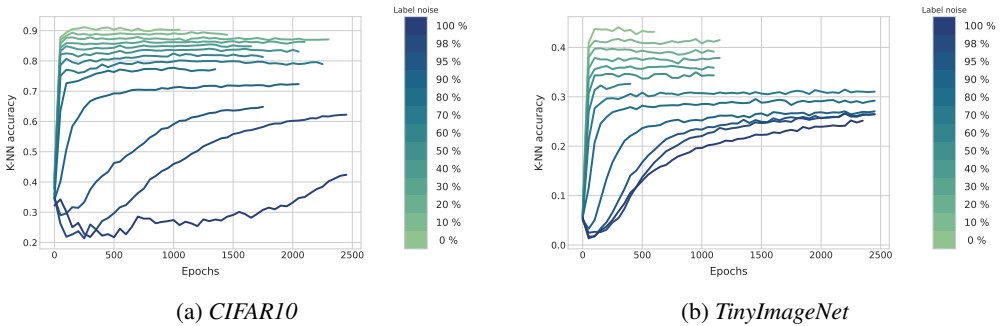

(a) *CIFAR10*                                    (b) *TinyImageNet*

Figure 10: Nearest-neighbour probing during training for the *CIFAR10* and *TinyImageNet* datasets during training.

**Effect on the strength of the augmentation.** Although malign memorization gets impossible as the number of augmentations keeps increasing, the strength of the augmentations themselves plays a role on the generalization achieved. The strength of the augmentation basically dictates the strength of the invariance that the model has to learn, which also correlates with the quality of the features learned as seen in Fig. 11. To control strength we use the strength level of RandAugment (Cubuk et al., 2020). We clearly see that stronger augmentations are beneficial for the quality of the embeddings, underlining the fact that invariance indeed plays a key role in benign memorization.

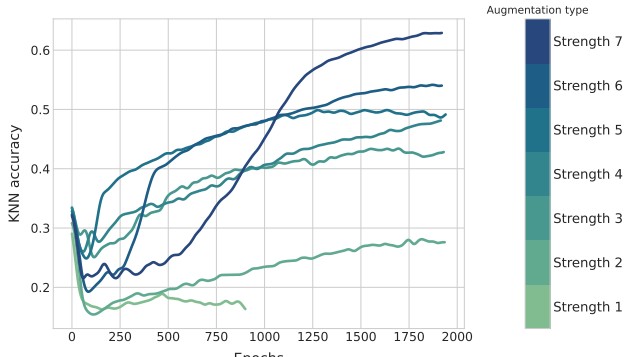

Figure 11: Nearest-neighbour probing accuracy when trained on random labels with varying strength of (infinite/online) augmentations.

**Varying levels of label noise.** Previous works have highlighted the importance of augmentations in the presence of high-label noise. We examine the relationship between label noise and generalization both with and without augmentations in Fig. 12. We see that while data augmentation is not so crucial for clean labels, its role becomes more and more critical as we increase the amount of label noise. Without augmentations, we suffer from a strong decrease in performance, eventually ending with random guessing as we reach complete label noise. On the other hand, using augmentations stabilizes this decay and we still achieve strongly non-trivial performance even in the setting of complete label noise.

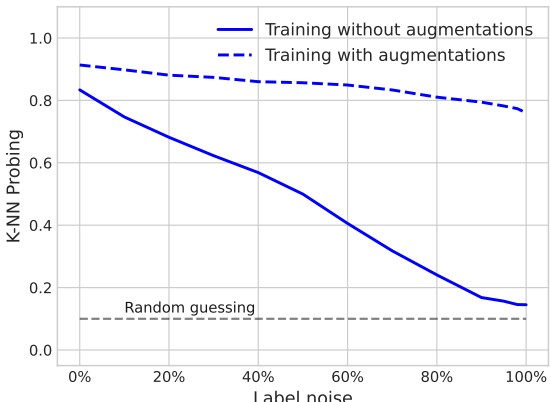

Figure 12: Nearest-neighbour probing accuracy when trained on varying levels of label noise with and without augmentations.

## D MORE DISCUSSION

### D.1 MODEL'S CAPACITY

Here we mathematically introduce the concept of capacity of a model class trained with Gradient Descent (GD) that was informally used in the main text. We remark that similar definitions have been explored in prior work (Arpit et al., 2017).

**Definition D.1** *Consider* $x_1, \ldots, x_n$ *for* $n \in \mathbb{N}$ *in general position and a fixed number of classes* $C$. *Given a labeling* $\tilde{\mathcal{S}} = \{(x_i, \tilde{y}_i)\}_{i=1}^n$, *let* $\mathcal{A}_t(\tilde{\mathcal{S}}) \subset \Theta$ *denote the set of solutions reachable by GD*

*with a computational budget $t$. We define the capacity $\mathcal{C}_t$ of the model class $\{f_{\boldsymbol{\theta}} : \boldsymbol{\theta} \in \Theta\}$ as*

$$\mathcal{C}_t := sup\{n \in \mathbb{N}| \; \forall \tilde{\mathcal{S}} \; \exists \boldsymbol{\theta} \in \mathcal{A}_t(\tilde{\mathcal{S}}) \; s.t. \; f_{\boldsymbol{\theta}} \; memorizes \; \tilde{\mathcal{S}}\}.$$

While similar to standard measures such as the VC-dimension (Vapnik & Chervonenkis, 1971), the capacity defined here depends on the learning algorithm (including the computational budget $t$) and is thus always smaller than the corresponding VC-dimension.

### D.2 RELATED WORKS

We give a more detailed discussion on the related work of Dosovitskiy et al. (2014) since although very different, a quick reading of it can mislead the reader into finding it more similar than is actually the case. The authors in Dosovitskiy et al. (2014) consider a dataset without labels, and sample $N$ patches from different images, leading to examples $\{\boldsymbol{x}_1, \ldots, \boldsymbol{x}_N\}$. $K$ augmentations are produced for each sample and all of those augmentations get the same label $i$. There are thus $N$ so-called surrogate classes. When varying the number of surrogate classes in Figure 3, the authors also adjust the number of samples used in the experiments, if the authors use 8000 surrogate classes, they also use 8000 patches, if they use 16000 classes they also use 16000 patches. At no point do different samples share the same label as the dataset size is not preserved in each experiment. This is in contrast to our experiment in Fig. 8 where we indeed vary the number of classes and assign the same labels to several samples. Having a label per sample of course strongly differs from only having 10 labels (or $C$, depending on the dataset). First, achieving strong performance in this setting is far more difficult (and thus surprising) since by reducing the number of classes so drastically, we are introducing a bias into the dataset as examples with different underlying labels might suddenly share the same label. We refer to the thought experiment in Sec. 5 for more details and remark that this reasoning does not apply for Dosovitskiy et al. (2014) precisely due to the fact that they use the same number of labels as samples. Second, our setup recovers the well-studied setting of random labels and thus memorization, connecting hence two very different fields, further distinguishing our results from Dosovitskiy et al. (2014) that were obtained in the context of self-supervised learning.

### D.3 INVARIANCE MEASURE

Here we give a bit more background and motivation for the invariance measure

$$I(\boldsymbol{x}; q, \mathcal{T}) := \frac{\mathbb{E}_{T_1, T_2 \sim \mathcal{U}(\mathcal{T})} \|q(T_1(\boldsymbol{x})) - q(T_2(\boldsymbol{x}))\|_2}{\mathbb{E}_{\boldsymbol{x}' \neq \boldsymbol{x}} \|q(\boldsymbol{x}) - q(\boldsymbol{x}')\|_2}.$$

This measure is inspired by loss functions often used in self-supervised learning where we aim to explicitly optimize for invariance to different augmentations (Chen et al., 2020c). We use a normalization term to ensure that there is a diversity between predictions of different, unrelated samples in order to rule out that simple collapse (i.e. constant) representations do not achieve a high invariance.

## E TOY EXAMPLE

To better understand the role of augmentations with respect to the capacity of the model we devise a simple toy setting.

We consider samples $\boldsymbol{x_i}, \ldots, \boldsymbol{x_n} \in \mathbb{R}^d$, that belong to one of the $C$ underlying classes. We select the first $d_1 < d$ coordinates to denote the true class assignment $z_i$, uniformly selected from the set $\{1, \ldots, C\}$, by sampling from a mixture of Gaussian distributions and let the rest $d - d_1$ dimensions correspond to noise. More specifically:

$$\boldsymbol{x_i}_{[:d_1]}|_{z_i=k} \sim \mathcal{N}(\boldsymbol{\mu}_i, \boldsymbol{\Sigma_1}),$$
$$\boldsymbol{x_i}_{[d_1:]} \sim \mathcal{N}(\boldsymbol{0}, \boldsymbol{\Sigma_2}).$$

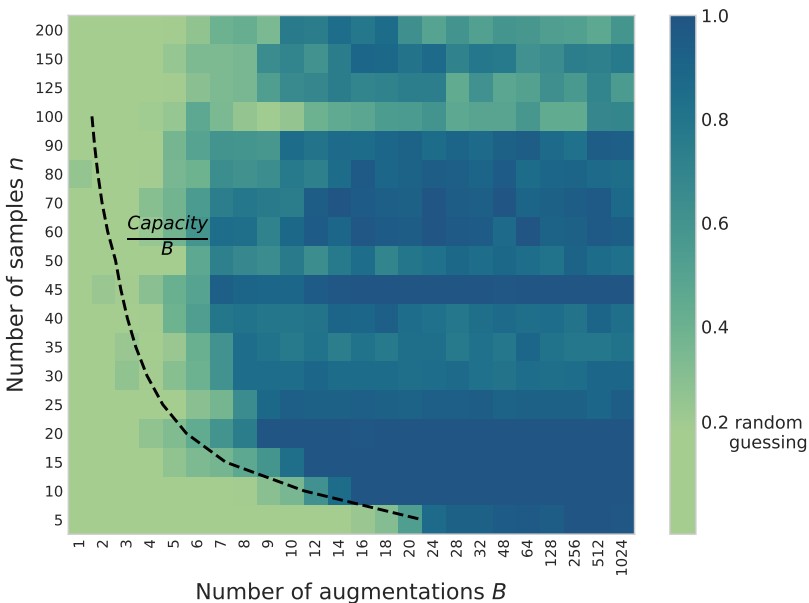

Figure 13: $K$-NN probing for the downstream class assignment on unseen test points for the toy example, after training on random labels.

With $\boldsymbol{\mu}_i$ we denote the cluster centers that are sampled randomly. In this case, we consider augmentations of the samples:

$$\bar{\boldsymbol{x}} = \boldsymbol{x} + \boldsymbol{a},$$
$$\text{where } \boldsymbol{a}_{[:d_1]} = \boldsymbol{0}$$
$$\text{and } \boldsymbol{a}_{[d_1:]} \sim \mathcal{N}(\boldsymbol{0}, \boldsymbol{\Sigma_3}).$$

As set the covariance matrices as $\Sigma_3 = \Sigma_2 = 10\,\Sigma_1 = \sigma^2 \boldsymbol{I}$, where $\boldsymbol{I}$ is the identity matrix. It is obvious that learning invariance for this task under these augmentations leads to embeddings that ignore the noise subspace. We employ in this case a simple linear encoder $h_{\boldsymbol{W}}(\boldsymbol{x}) = \boldsymbol{W}\boldsymbol{x}$, with $\boldsymbol{W} \in \mathbb{R}^{d \times d}$.

As a projector we use $g_{\boldsymbol{V}}(\boldsymbol{x}) = \sum_{i=1}^{K} \dfrac{\frac{1}{\|\boldsymbol{x} - \boldsymbol{v_i}\|}}{\sum_{j=1}^{K} \frac{1}{\|\boldsymbol{x} - \boldsymbol{v_j}\|}} \boldsymbol{l_i}$. Here $\boldsymbol{l_i}$ is a one hot encoding sampled randomly from the set $\{\boldsymbol{e_1}, \ldots, \boldsymbol{e_C}\}$. We choose such a projector as it is not difficult to verify that in the limit case where $g_{\boldsymbol{V}}(\boldsymbol{x}) = \boldsymbol{l}_{\arg\min_i \|\boldsymbol{x} - \boldsymbol{v_i}\|}$, we have an upper bound on the capacity of the model, based on the number $K$ of the vectors used by the projector. Additionally, its non-linear nature proposes that any possible invariance learning will occur at the encoder. In Fig. 13 we visualize the $K$-NN probing for the downstream task of correctly predicting the clean label of unseen test samples, after training on random labels. By varying the number of samples $n$ and augmentations $B$ available, we directly control whether full memorization can take place or not. When the number of samples $n$ is smaller than $\frac{\mathcal{C}}{B}$, where $\mathcal{C}$ is the capacity of the model, full memorization is possible, which discourages any feature learning, leading to bad generalization.

## F    EXPERIMENTAL SETUP

**Dataset Details:**   We conducted experiments on the classic *CIFAR-10*, *CIFAR-100* and *TinyImageNet* datasets, using the default dataset splits. For completeness, we present in Table 5 statistics regarding these datasets.

| Dataset | Examples in train split | Examples in test split | Number of classes |
|---|---|---|---|
| *CIFAR-10* | 50000 | 10000 | 10 |
| *CIFAR-100* | 50000 | 10000 | 100 |
| *TinyImageNet* 3 | 100000 | 10000 | 200 |

Table 5: Statistics for the datasets used.

**Architecture:**  As commonly done for smaller size images (Chen & He, 2021; Hua et al., 2021a), we use a variant of the *ResNet18* architecture, where the max-pooling layer is removed, and the first convolution is modified to have a kernel size of 3 and a stride of 1. We also remove the last fully connected layer, as we replace it with our projector. We provide more details about the hyperparameters used in Table 6. For the experiments on *TinyImageNet*, we rescale images to a size of $64 \times 64$ and additionally restore the stride of the first convolution to the value 2. We apply the same modification to VGG when trained on *TinyImageNet*.

| | **Hyperparameters** | **Value** |
|---|---|---|
| Augmentations | Random cropping scale | (0.08, 1) |
| | Horizontal flip probability | 0.5 |
| | Color-Jittering | (0.8, 0.8, 0.8, 0.2) |
| | Grayscale probability | 0.2 |
| | Mixup | yes |
| Parameters | image-size | 32/64 |
| | clip-norm | 'none' |
| | dropout | 'none' |
| | Projector size | 65536 |
| | Projector MLP normalization | 'none' |
| Training | Learning rate | $4\epsilon^{-3}$ |
| | Learning rate scheduler | 'none' |
| | Adam $(\beta_1, \beta_2)$ | (0.9, 0.999) |
| | Batch size | 256 |
| | Weight decay | 0.0 |
| | Loss | 'MSE-loss' |

Table 6: Hyperparameters for the random labels experiments.

