# OpenReview forum: "The Curious Case of Benign Memorization"
_ICLR.cc/2023/Conference — ICLR 2023 poster_

### Official Review · Reviewer_ZaD4 · 2022-10-19

**Confidence:** 4
**Correctness:** 3
**Technical Novelty And Significance:** 3
**Empirical Novelty And Significance:** 3
**Recommendation:** 6

**Clarity, Quality, Novelty And Reproducibility:**

Overall this work is clearly organized, with high-quality experiments and novel designs.

I could not check the reproducibility since there are no supplementary materials or anonymous github links. I hope the authors could release the code.

**Strength And Weaknesses:**

Strength:
1. The paper is well-written and easy to follow. The idea is well-motivated from recent benign overfitting papers.
2. The discovered phenomenon is very interesting and provides useful insights to theory works.

Weakness:
1. How does the augmentation magnitude affect the results? In my experience, if the augmentation is too strong, the network cannot fit the data.
2. I am curious about what role does BatchNorm play in this study. Current theory works (at least for benign overfitting) do not consider BN. I would appreciate it if the authors could verify if BN is important to helping early layers learn useful features under random labels.
3. In addition to K-NN, will training a linear classifier on fixed feature mappings (learned from random labels) also exhibit similar results?
4. Related feature learning behavior is also mentioned in previous application papers, just name a few [1]. Basically, people also found the network backbone is responsible for feature learning and the classifier is more prone to label quality. The authors may consider discussing more on these related works.

[1] To Balance or Not to Balance: A Simple-yet-Effective Approach for Learning with Long-Tailed Distributions

**Summary Of The Paper:**

This work discovers interesting phenomena about benign overfitting: 1) even training on random labels, early layers can still learn meaningful representations; 2) deep layers are prone to fit noises in data and cannot learn useful features.

**Summary Of The Review:**

In general, I think this is a good paper meaningful to both theory and empirical researchers.

---

> ### Author Response · Authors · 2022-11-09
> **Response to Reviewer ZaD4**
>
> We thank the reviewer for the interesting feedback and suggestions. We are happy to hear that our results are meaningful for both theoretical and empirical research.
>
> 1. **Strength of Augmentations.** Thank you for the interesting question. We assume that by *strength*, the reviewer refers to the degree to which the transformations are applied to the input images while still preserving the properties of the data distribution (for instance the rotation angle).  We have performed experiments where we vary the degree of strength of the augmentation using the framework of RandAug, see Figure 11 in the Appendix. We find that stronger augmentations generally help performance in this setup. Somewhat unsurprisingly — and in agreement with the reviewer — we have also observed that more diverse augmentations lead to longer convergence time, as the model needs to learn more complicated invariances.
> On the contrary, if the strength indicates the degree to which the images are distorted (for instance too small crops or too strong jittering), the resulting embedding will not generalize when performing linear probing or $K$-NN.
> 2. **BatchNorm.** Thank you for the suggestion, we are currently experimenting with architectures that exclude the usage of batchnorm. We will report the outcome of this experiment once it is completed.
> 3. **Probing.** We refer the reviewer to the comment addressed to all the reviewers. We find only a marginal difference between $K$-NN and linear probing.
> 4. **Code and details.** For training details and experimental setup we refer to Appendix F (revised paper). We have further attached our code to the supplementary material.

---

### Official Review · Reviewer_rsqh · 2022-10-21

**Confidence:** 4
**Clarity, Quality, Novelty And Reproducibility:** See above.
**Correctness:** 3
**Technical Novelty And Significance:** 2
**Empirical Novelty And Significance:** 2
**Recommendation:** 5

**Strength And Weaknesses:**

The topic of memorization and generalization in deep learning is of high relevance to the ICLR community. The paper is mostly well-written (apart from some parts references below) and the details for reproducing the work are provided. The main findings described above are supported convincingly.

However, I think the paper has limited technical and experimental novelty. Furthermore, the most related work of Dosovitskiy et al. [3] is not discussed in the main text. It is referenced only in the appendix, in the context of an experiment (Fig. 8) which is essentially a reproduction of the experiment described in section 4.3.1 of Dosovitskiy et al. [3].

1. The finding that it is possible to learn meaningful representations of data by fitting random labels/targets was well-established by Dosovitskiy et al. [3], where a neural network is trained with random surrogate classes (in the limit each example gets its own class) and **data augmentations**. It is clear that it is the presence of data augmentations that provides the signal for learning useful representations. Dosovitskiy et al. [3] also analyze the effect of the number of the surrogate classes.

2. The finding that label noise memorization happens mostly in later layers of neural networks has been presented before (see for example Arpit et al. [1] and Cohen et al. [2]).

3. The observation that training with data augmentations leads to representations that are more invariant to data transformations is also present in Dosovitskiy et al. [3]. I believe this direction is still underdeveloped. Exploring more the theoretical aspect of this (along the lines of arguments presented in the appendix A) and understanding what kind of representation properties emerge from different types of data augmentations can significantly improve this work. For example, it would be interesting to find out what is the exact effect of the MixUp augmentation on representations?


### Minor comments

* I suggest to state early in the paper that the term “memorization” refers to the ability of memorizing a noisy label. In some existing works memorization is defined as the difference between training and test accuracies (mostly in the noiseless setting). Feldman & Zhang (2020) define it differently. Different settings and definitions of memorization will probably lead to different conclusions whether generalization and memorization are in conflict or not.

* “While previous works have identified that the very first layers provide non-trivial performance even without data augmentation, including it strengthens this effect significantly.“ – it would be nice to provide some citations here.

* I think the thought experiment presented in Sec 5.1 can be removed. If one aims to check whether the label-preserving property of data augmentations is the key that enables learning of useful representations, then the natural experiment to do is to generate a random label with every augmentation and check whether learning still happens. This is indeed done in Figure 5 and clearly demonstrates that the signal comes from many similar examples having the same [random] label.

* The normalized invariance of eq. (2) is defined using predictions of the network. It is expected that this invariance score is going to be lower when training with data augmentations, as essentially the network is supervised to output the same target for all transformations of one example. I think it will be more appropriate to probe *representations* in order to check if representations become more invariant to data transformations.

* Typos to fix: “totice” → “notice” and ““applied3” → “applied”.

* In Figure 6, the dashed curve values can be presented on the right y-axis.

* The main findings of the experiments presented in the appendix can be summarized concisely in the main text.

### References

[1]  Devansh Arpit, Stanisław Jastrzebski, Nicolas Ballas, David Krueger, Emmanuel Bengio, Maxinder S Kanwal, Tegan Maharaj, Asja Fischer, Aaron Courville, Yoshua Bengio, et al. A closer look at memorization in deep networks. In ICML, 2017.

[2] Gilad Cohen, Guillermo Sapiro, and Raja Giryes. DNN or k-NN: That is the generalize vs. memorize question. ArXiv, abs/1805.06822, 2018.

[3] Alexey Dosovitskiy, Jost Tobias Springenberg, Martin Riedmiller, and Thomas Brox. Discriminative unsupervised feature learning with convolutional neural networks. In NeurIPS, 2014.



**Summary Of The Paper:**

This submission considers neural networks trained with completely random labels with or without data augmentations. The main observation is that when data augmentations are enabled, neural networks learn meaningful representation of data. This is supported by fitting a k-NN classifier on learned representations using ground-truth labels, and demonstrating significantly increased performance compared to representations at initialization. The explanation that the authors give for this phenomenon is that data augmentation increases the effective dataset size, which given the limited capacity of the neural network, makes it impossible to memorize random labels without uncovering the structure in the augmented data. The paper also highlights that the label noise memorization usually happens in later layers of neural networks.


**Summary Of The Review:**

In summary, I think the findings of this submission are not novel enough to be published at ICLR. This is the main determinant of my recommendation.

UPDATE: Given that I had misunderstood the setting of Dosovitskiy et al. [3] initially, after the discussion with the authors, I have updated my recommendation from 3 to 5.

---

> ### Author Response · Authors · 2022-11-09
> **Response to Reviewer rsqh (Part I)**
>
> We thank the reviewer for the constructive review and suggestions. We will address the questions and comments in the following, with an emphasis on the novelty of our work.
>
> 1. **Novelty of random labels.** We agree that [3] indeed is related as each sample is assigned its own label and augmentations are applied, but we want to clarify that their considered setting is **very different** from ours. We remark that even in their *varying the number of surrogate classes* experiments, still every sample gets its own label and there is no label-sharing between different examples, making their experiments significantly different from ours. For full clarity, we will describe the setup considered in [3] and contrast it with ours. We realize that we should have clarified this and apologize for the confusion. We have added [3] to the related works section and further provide a detailed description of the paper (Appendix D1), clarifying the differences to our work. Moreover, we would be very happy to further discuss with the reviewer if he/she disagrees with our reading of [3]. \
> The authors in [3] consider a dataset without labels, and sample $N$ patches from different images, leading to examples {$x_1, \dots, x_N$}.
> $K$ augmentations are produced for each sample and all of those augmentations get the same label $i$. There are thus $N$ so-called surrogate classes. When varying the number of surrogate classes in Figure 3, the authors also adjust the number of samples used in the experiments, if the authors use 8000 surrogate classes, they also use 8000 patches, if they use 16000 classes they also use 16000 patches. **At no point do different samples share the same label** as the dataset size is not preserved in each experiment. This is in contrast to our experiment in Figure 8 where we indeed vary the number of classes and assign the same labels to several samples. \
> Having a label per sample of course strongly differs from the setting that we consider where we only have 10 labels (or $C$, depending on the dataset). Firstly, achieving strong performance in this setting is far more difficult (and thus surprising) since by reducing the number of classes so drastically, we are introducing a bias into the dataset as examples with different underlying labels might suddenly share the same label. We refer to the thought experiment in Section 5 for more details and remark that this reasoning does not apply for [3] precisely due to the fact that [3] uses the same number of labels as samples. Secondly, our setup recovers the well-studied setting of random labels and thus memorization, connecting hence two very different fields, further distinguishing our results from [3] that were obtained in the context of self-supervised learning.
> 2. **Novelty of memorization trade-off.** We agree with the reviewer that the discrepancy in memorization between early and late layers has been pointed out in the literature, as we also mention in the main text. We however remark that our results are **significantly stronger**: [1, 2] show that only the last layers can fit the random (training) labelling based on simple $K$-NN or linear probing. In contrast, we show a stronger difference in the sense that under data augmentation, early layers can reach very strong test performance, while the last layers do not generalize at all (and thus truly memorize). Such a discrepancy is not visible in [1] and [2] and marks a crucial distinction between early and late layers.
> 3. **Importance of augmentations.** We agree that the importance of augmentations in self-supervised/unsupervised settings has been remarked in several previous works, including [3], as we highlighted in the main text. The significant difference that data augmentation can make in a random labelling setting however **has not** been observed to the best of our knowledge. As argued in point 1. and in the thought experiment in Section 5, such a setting is significantly more challenging than the one in [3].
> 4. **Definition of memorization.** We completely agree with the reviewer and we will add a precise statement regarding what exactly we mean with memorization, i.e. perfectly fitting the random training labels. We then further distinguish between malign memorization (i.e. failing to learn useful features for clean test data) and benign memorization (i.e. learning useful features for clean test data). Such a definition might also better highlight our novel contributions in terms of point 2 where usefulness in terms of clean test data has not been assessed.
> 5. **Citation.** Thank you for pointing this out. [4] has shown that training from a checkpoint obtained by pretraining with random labels can sometimes lead to faster convergence for the clean task. We have rephrased the text accordingly and added the citation.

---

> > ### Author Response · Authors · 2022-11-09
> > **Response to Reviewer rsqh (Part II)**
> >
> > 6. **Thought experiment.** We believe the thought experiment is indeed useful as it shows something stronger than Figure 5. Merely preserving the label is not enough in the random label setting, for instance when using i.i.d. augmentations, the underlying label is also preserved but such augmentations are not useful as can be seen in Table 3. The correlation between two augmentations that preserve the label is also essential, an insight that cannot be gained from the experiment in Figure 5.
> > 7. **Invariance measure.** Thank you for pointing this out, we have indeed assessed the invariance of the representations but wrongly defined the invariance in terms of the output of the model. We have corrected this mistake in the new version.
> >
> > Given that the reviewer’s decision was mainly based on the lack of novelty of our contribution and in light of our clarifications on that matter, we would greatly appreciate it if the reviewer could re-assess his/her evaluation and consider raising the score. We are happy to further clarify remaining doubts.
> >
> > [1] Devansh Arpit, Stanisław Jastrzebski, Nicolas Ballas, David Krueger, Emmanuel Bengio, Maxinder S Kanwal, Tegan Maharaj, Asja Fischer, Aaron Courville, Yoshua Bengio, et al. A closer look at memorization in deep networks. In ICML, 2017.
> >
> > [2] Gilad Cohen, Guillermo Sapiro, and Raja Giryes. DNN or k-NN: That is the generalize vs. memorize question. ArXiv, abs/1805.06822, 2018.
> >
> > [3] Alexey Dosovitskiy, Jost Tobias Springenberg, Martin Riedmiller, and Thomas Brox. Discriminative unsupervised feature learning with convolutional neural networks. In NeurIPS, 2014.
> >
> > [4] Maennel, Hartmut, et al. "What do neural networks learn when trained with random labels?." Advances in Neural Information Processing Systems 33 (2020): 19693-19704.

---

> > > ### Comment · Reviewer_rsqh · 2022-11-15
> > > **Response to the authors**
> > >
> > > Thank you for the detailed response.
> > >
> > > #### **On the connectinos with Dosovitskiy et al. [3].**
> > > Taking multiple random patches from one image can be seen as a part of data augmentations (random crops).
> > > Therefore, we can safely conclude that the main setting of Dosovitskiy et al. [3] is the same as the "per-sample" setting considered in this work (Figure 8), where each original image receives its own label.
> > > Indeed, the authors are right that changing the number of surrogate classes in Dosovitskiy et al. [3] is not the same as changing the number of classes in this paper.
> > > I will update my review on this after the correspondace with the authors.
> > >
> > > In the light of [3], the central finding of this work is than when one partitions the $n$ surrogate classes (one per example) into $C$ groups/classes randomly, learning of useful features still happens, unless $C$ is too small (see for example $C=2$ of Figure 8).
> > > The authors claim that this parititioning of surrogate classes into $C$ classes somehow creates a more difficult learning problem.
> > > > Having a label per sample of course strongly differs from the setting that we consider where we only have 10 labels (or $C$, depending on the dataset). Firstly, achieving strong performance in this setting is far more difficult (and thus surprising) since by reducing the number of classes so drastically, we are introducing a bias into the dataset as examples with different underlying labels might suddenly share the same label.
> > >
> > > One one hand, empirically we see that this is true (Figure 8).
> > > One the other hand, all that was changed was partitioning the surrogate classes into $C$ groups, which a neural network can do too with an extra linear layer at the end.
> > > It is not clear why we observe a shift from benign memorization to malign memorization as one decreases $C$.
> > > To me it seems that this is the crux of the paper and most of the effort should have been spent on exploring this.
> > > I suggest to emphasize this aspect more.
> > >
> > > Note that the capacity argument does not explain this, as reducing $C$ does not increase the effective training set size, thus causing malign memorization.
> > > Similarly, the discussion in Appendix B does not provide any answers.
> > > When decreasing $C$, the bias term will push averages of predictions (over data augmentations) of different images (that possibly have different true labels) towards the same one-hot vector.
> > > It is not clear to me how this causes malign memorization.
> > >
> > > #### **On label-preservation**
> > > I still find the "label-preservation" discussion of Section 5 a bit confusing.
> > > I think part of the confusion comes from the term "label-preservation", as its unclear whether the authors mean preservation of the true label or the assigned/random label. I assume the latter.
> > >
> > > In my deconstruction of data augmentation, it is (a) creating variations of one iamge and (b) keeping the *assigned* label the same. The thought experiment is a case where the property (b) is kept but property (a) is not respected (by generating fresh examples that share the same true label).
> > > The failure to learn usuful features in this setting does not mean that assigned-label-preservation is not important.
> > > It means that the property (a) was important and that property (b) alone is not enough. In fact, both (a) and (b) play an important role.

---

> > > > ### Author Response · Authors · 2022-11-16
> > > > **Response to the reviewer**
> > > >
> > > > Thank you for the detailed response and enabling an interesting discussion.
> > > >
> > > > **Number of surrogate classes:** We are glad to hear that the reviewer agrees that the varying surrogate classes experiment is not the same as our setting. We agree with the reviewer that the limiting case where the number of classes matches the number of samples, reproduces the setting of [3]. We remark however that we only considered such a setting in Figure 8 and mainly focus on the case where the number of classes matches the original dataset, as this is the commonly studied setting in the memorization/random labels literature.\
> > > > **Difficulty of learning less classes:** We want to emphasize once more the differences of our work compared to the setting in Dosovitskiy et al. [3]. In their experiments no memorization setting in the classic sense of random labels is studied, as each sample is assigned its own class. We agree with the reviewer that in principle, learning an additional linear layer on top would suffice to learn any random grouping, provided that the network first learns a per-sample representation. It is however entirely unclear if a network adopts such a strategy as it does not receive per-sample information but rather conflicting labels that it has to memorize. Even if that were the case, this would remain highly surprising. \
> > > > **Shift to malign memorization:** The number of classes $C$ in this case, controls the amount of noise we insert in the process. We note that using fewer classes (e.g. $C=2$) in Figure 8, does not lead to malign memorization as the features still enable non-trivial generalization. Excluding data augmentation however does lead to malign memorization as the learnt features have no generalization power. Hence, there is no shift to malign memorization as $C$ is decreased.
> > > > We also note that our capacity argument does not guarantee benign memorization. It just rules out complete malign memorization. Notice that the capacity argument does not aim at explaining *how* benign memorization occurs, but rather that it needs to occur.\
> > > > **Augmentations:** Throughout the paper we use augmentations to denote the process of generating two views $(T_1(\boldsymbol x), \boldsymbol y)$, $(T_2(\boldsymbol x), \boldsymbol y)$ from an original datapoint $(\boldsymbol x, \boldsymbol y)$, where $T_1, T_2 \sim \mathcal{U}\left(\mathcal{T}\right)$. Note that the label $\boldsymbol y$ in this case may be the clean or the random one. Hence, in both cases the augmentation point preserves the assigned label. In Section 5, we take the set of transformations $\mathcal{U}\left(\mathcal{T}\right)$ to the extreme, where we return a new image from the original dataset based on the true label of the original sample. This is done to highlight the importance of data augmentation.
> > > > We thank the reviewer for providing more intuition and an explanation about the role of augmentations. We remark that the interpretation provided is in-line with our arguments, also highlighted in the text. We quote from the Section 5:
> > > >
> > > > *This thought experiment demonstrates that under random labels, augmentations seem to play a very distinct role compared to the clean setting and label preservation in itself is not enough to guarantee benign memorization. This raises the following question:*
> > > >
> > > > *What other properties of augmentations, besides label preservation, cause benign memorization?*
> > > >
> > > > *We hypothesize that the origins of the phenomenon lie at the interplay between the highly correlated nature of augmentations and the inflated effective sample size that exceeds the model capacity (Sec 5.1), forcing the model to learn meaningful features (Sec 5.2).*
> > > >
> > > > We hope that our answer clarifies some of the concerns of the reviewer and are happy to elaborate if something remains unclear.

---

> > > > > ### Comment · Reviewer_rsqh · 2022-11-17
> > > > > **Response to the authors**
> > > > >
> > > > > Thank you for the follow-up.
> > > > >
> > > > > > We note that using fewer classes (e.g. $C=2$) in Figure 8, does not lead to malign memorization as the features still enable non-trivial generalization.
> > > > >
> > > > > The performance at the end for $C=2$ is almost the same as at initialization. According to the Definition 4.1, this is malign memorization.
> > > > >
> > > > > > We also note that our capacity argument does not guarantee benign memorization. It just rules out complete malign memorization. Notice that the capacity argument does not aim at explaining how benign memorization occurs, but rather that it needs to occur.
> > > > >
> > > > > That was clear from the text. My main point was that the arguments provided in the paper (the role of data augmentation and capacity) do not explain the declining performance when $C$ decreases.
> > > > >
> > > > > **Summary**
> > > > >
> > > > > I understand that the authors focus on the case where the number of classes of random labels ($C$) is the same as the number of dataset classes ($C^*$). Given that it was shown that setting $C=n$ leads to learning of useful representations, and that this paper focuses on the $C=C^*<n$ case, the paper should have focused on the following:
> > > > >
> > > > > (a) demonstrating that these cases are different, and
> > > > >
> > > > > (b) explaining why these cases are different.
> > > > >
> > > > > The former was addressed partly, although it was not highlighted as a central result. The latter was not addressed. I agree with the authors that, intuitively, decreasing $C$ creates some "conflicting" labels, but this cannot be regarded as a proper explanation.
> > > > >
> > > > > Given that I had misunderstood the setting of Dosovitskiy et al. [3] initially, I have updated my recommendation from 3 to 5.

---

> > > > > > ### Author Response · Authors · 2022-11-18
> > > > > > **Response to Reviewer**
> > > > > >
> > > > > > We thank the reviewer for engaging in the discussion and raising the score.
> > > > > >
> > > > > > We agree that further exploration of the regime of varying classes is a very interesting direction and warrants further investigation.

---

### Official Review · Reviewer_dZQx · 2022-10-24

**Confidence:** 4
**Correctness:** 3
**Technical Novelty And Significance:** 2
**Empirical Novelty And Significance:** 3
**Recommendation:** 6

**Clarity, Quality, Novelty And Reproducibility:**

The paper is clearly written and easy to follow. Experimental details are provided in the appendix, and the experiments seem to be relatively easy to reproduce. I do have the following questions:

1. Can you clarify how the "Non-label preserving augmentation" are generated?

2. Can you explicitly clarify in the main text how "training loss" are defined and computed under data augmentation? Do you simply report the running average of the training loss, which is the loss computed *before* the model trains on each augmented version of the image? Or do you load a model checkpoint and go over the entire training set + augmented copies and compute the average loss?

**Strength And Weaknesses:**

**Strength**

1. This paper identifies an interesting phenomenon called benign memorization, which has not been systematically studied in the past.
2. This paper analyzed this phenomenon from various aspects, including identifying where memorization happens, and potential explanation from the perspective of effective model capacity.

**Weakness**

1. The experiments in Section 5 with i.i.d. data aug are a bit puzzling, and would benefit with more in-depth analysis. For example, to what extent is the observation due to the reduced base training set size (1000 vs 50,000)? Can you include 2 extra rows in Table 3 for Random + standard DA, and Clean + standard DA (potentially also compare to online DA vs. 50 copies of pre-generated augmentations).

2. I'm a little confused by the arguments from effective capacity under a fixed computational budget. The paper argues that the models do not have enough capacity to efficiently memorize with augmented samples, therefore it is forced to benign memorization.

    1. The argument does not explain why the models do not do benign memorization (i.e. learning good representations and only memorizing with the projector) when the capacity is enough.
    2. The main phenomenon of benign memorization is not under a computational budget. The model in Fig.3 was trained for 13,000 epochs and still demonstrate benign memorization. Does the effective capacity argument help with the understanding here? One potential argument is that this experiment is done with online augmentation, leading to increasingly larger training set size as training goes. To complete the picture, it would be good to conduct the same experiment (i.e. *without* computation budget) but with fixed number of standard data augmentations.



**Summary Of The Paper:**

This paper studies a notion coined as "benign memorization", which refers to the phenomenon that when training with standard data augmentation, a deep neural network could fit to complete random label assignment on the training set, yet still learns representations that shows surprisingly good discrimination power under kNN probe (using the *clean* training and test labels). The paper went on to conduct a number of experiments aim to understand this behavior, and found that different layers behave differently, and the phenomenon cannot be fully explained by the label-preserving property of data augmentation.

**Summary Of The Review:**

This paper identifies an interesting phenomenon of model learning representation with surprisingly good discriminative power even for completely random label assignment, when trained under standard data augmentation. This could potentially provide an interesting angle for future studies of the role of data augmentation in deep learning, and in particular, in self-supervised learning.

---

> ### Author Response · Authors · 2022-11-09
> **Response to Reviewer dZQx**
>
> We thank the reviewer for the precise reading of our work and the suggestions made. We are glad to hear that our work might provide a novel angle for future studies of augmentations in deep learning. In the following, we will address the questions and comments.
>
> 1. **Number of I.I.D. Augmentations.** Thank you for the suggestions, it indeed is very insightful to check if non-i.i.d. (i.e. standard) augmentations can improve over random guessing if only 1000 datapoints are used, both under random and clean labels. We have added the rows to Table 3 and find indeed that both settings perform non-trivially in contrast to the i.i.d. setting, further highlighting that the correlated nature of standard augmentations is crucial.
> 2. **Capacity.** We agree with the reviewer that the capacity argument can only rule out malign memorization above the capacity threshold, but does not allow for any conclusions when we are below the threshold (i.e. whether we malignly or benignly memorize). Given however that in standard settings data augmentation is applied in an online fashion (leading to infinitely many samples), we are operating above this limit, and our capacity argument hence applies and rules out the possibility of malign memorization. Still, understanding what happens below the capacity limit remains very interesting and we leave further exploration for future work.
> 3. **Fixed number of augmentations.** We have performed experiments with a fixed number of augmentations in Figure 5. We find that malignly memorizing the augmentations (i.e. randomizing the label for each augmentation) becomes infeasible for a certain number of augmentations (orange curve in 5b) and computational budget, while benign memorization emerges more and more as we increase the number of augmentations (blue curve in 5a). Our capacity argument thus also applies in a non-online setting.
> 4. **Non-label preserving.** We generate augmentations and assign a random label to each augmentation, i.e. two augmentations of the same image do not necessarily share the same label.
> 5. **Training loss.** We report the running loss incurred before training on the specific samples. We have updated the text to reflect that.

---

> > ### Comment · Reviewer_dZQx · 2022-11-18
> > **Reply to the authors**
> >
> > Thanks for the authors for the clarification and additional results. This clarifies most of my questions, thus I'm keeping my positive rating.

---

### Official Review · Reviewer_CAZe · 2022-10-25

**Confidence:** 4
**Correctness:** 3
**Technical Novelty And Significance:** 3
**Empirical Novelty And Significance:** 4
**Recommendation:** 8

**Clarity, Quality, Novelty And Reproducibility:**

* Clarity: The paper was for the most part clear, though I have mentioned a few minor points below:
    * pg. 1: "commands new avenues" -- commands is odd word choice, perhaps "suggests" is better
    * pg. 1: typo "applied3"
    * More clarity about the train-test splits for the $k$-NN procedure should be included somewhere, might have missed it
    * Generally, the kind of data augmentation being used for random labels (i.i.d., non-i.i.d. procedure given in Section 4, etc) should be better clarified throughout the paper, I was confused several times about which kind of data augmentation was being referred to. For example, in Table 2, in Fig. 5, in the "Learn if you must" section on pg. 8, and in the body of the paper throughout.
    * In Figure 3, consider using a histogram for the probing results graphs or instead indicating on the x-axes some arrow that signifies "increasing depth".
    * pg. 7: I wouldn't use the phrase "counter-intuitively", it seems to me at least pretty intuitive and expected that label-preserving i.i.d. augmentations for random labels would be bad.
    * pg. 8: "As we increase B", B should be in LaTeX format.
    * pg. 9: Fig. 6 should clarify if normalized invariance is averaged over the whole dataset and whether the $k$-NN probing error is using the embedding layer. It should also clarify what kind of data-augmentation is being used.
    * pg. 9: Eq. 2 doesn't appear to define which norm is being used (I assume $\ell_2$).
    * pg. 14: I believe Lemma A.2 is standard and that should be mentioned.
    * pg. 16: first word, "classes," -- there should be no comma there.
    * pg. 16: "recovering the result" -- it should be clarified what this result is, the sentence is unclear.
    * pg. 17: "RandAugment" -- it would be helpful to describe what strength means here.
    * pg. 17: "Previous works have highlighted the importance of augmentations" -- cite them
    * pg. 18: "We employ is this case a simple linear encoder" -- typo
    * pg. 18: In the Toy Example section, I was confused by the choice of projector and the justifications given for it -- what is the capacity of the model? Capacity in what sense? VC-dimension? I was also confused by the phrase "its non-linear nature guarantees any possible invariance learning will occur at the encoder instead" -- is that true? Why? This seems like a plausible hypothesis, but it's unclear that it's true. More description of where this particular projector comes from would also be helpful.
    * pg. 18-19: In Figure 13, how is the capacity $\mathcal{C}$ of the model computed (as in what formula are you using and why)?



* Quality: The paper's development and experiments are of high quality.


* Novelty: The paper has some very interesting (and to my knowledge, novel) experiments and insights.


* Reproducibility: The paper seems easy to try to reproduce, though I have not done so myself.

**Strength And Weaknesses:**

* Strengths:
    * This paper finds a very interesting set of phenomena regarding memorization in deep neural networks, and performs several thorough and well-thought-out experiments to delineate and explain the phenomena.
    * The paper empirically demonstrates that the embeddings learned under random noise are useful, and memorization happens in the last layer in their setting.
    * The paper empirically demonstrates the usefulness of augmented labels (and the added benefit in the presence of larger amounts of label noise) for the purpose of learning the useful (under $k$-NN) embeddings, and show that without augmentation, the embeddings learned are not useful (under $k$-NN).
    * The paper proposes an explanation for the usefulness of data augmentation that derives from a concept of invariance under augmentation which correlates with improved performance, and empirically shows that the embedding layer exhibits the most invariance.
    * The paper empirically demonstrates that larger sizes of (appropriately) augmented datasets results in higher difficulty in overfitting during training, and proposes this empirical fact as an explanation for why data-augmentation helps embedding learning.


* Weaknesses:
    * Only one relatively simple architecture for image classification is studied in the paper, and it is unclear whether the phenomena exhibited here will generalize to other kinds of architectures.
    * The heuristic arguments regarding the explanation of data augmentation's advantage in terms of capacity are rather loose and should probably be framed differently. In particular, I feel that Definition $5.1$ and the language of VC-dimension etc. are unnecessary and not useful framings to make the argument that empirically we see that with data augmentation, the training loss after 5000 epochs (Fig. 5b) no longer goes to zero (also, what do plots like Fig. 5b look like if we increase 5000 to say 20000?). Instead of giving a technical definition that is essentially a stand-in for whether or not zero training error is reached or not, it would be simpler and clearer to avoid giving that definition since it is not used anywhere else. I was also unclear about the phrase "as effectively $B \to \infty$" in the Inflated Sample Size section on page 8 -- this is also used throughout the paper, for instance in the description of Fig. 11, and it doesn't seem to be entirely correct to me.
    * Quality of embeddings is assessed solely via $k$-NN probing -- I agree this approach is very reasonable and interesting to study, and has indeed been used in many prior works. I would also be interested in the results of other forms of probes (say, training a linear classifier on top of the embeddings on true data) beyond just $k$-NN. It is also not entirely clear to me what the quality of embeddings with respect to $k$-NN probes implies about the quality of embeddings used in other ways (if this has been addressed in prior literature, perhaps this could be elaborated on in the paper or at least mentioned). For instance, it would be helpful to elaborate upon "due to the very restricted complexity of probing, the resulting performance is very dependent on the quality of input representations" (pg. 3). In particular, what complexity? Restricted in what way? Is quality being used loosely here? It would also be nice if pointers were included to any existing literature on measuring the quality of representations beyond $k$-NN probes (perhaps in the appendix).
   * The normalized invariance measure introduced in Eq. 2 seems reasonable, but it would be helpful to remark upon/ explain why this particular measure was chosen; in particular it is not clear to me that the value for each data point $x$ is on the same scale since there is included in the normalization a dependence on the particular dataset as well. This doesn't seem like too big of an issue since the measure seems to be averaged over the data points $x$ anyways (is this true? Fig. 6 wasn't clear on this), but some more discussion would be good to have.
   * Regarding related work, it might be a good idea to add more and more recent citations (later than 2019) to the section in the intro on progress in deep learning, benign overfitting, and the section on data augmentation (https://arxiv.org/abs/2106.04156 for instance among others for data augmentation). There are also of course several more works that are related in the memorization setting (other papers by Feldman for example). I would also particularly recommend the authors look at the line of work relating to understanding feature learning in deep networks (for instance, https://arxiv.org/abs/1905.10337, https://arxiv.org/abs/2001.04413 and prior citations as well as work that cites this paper), as it relates to some of the questions raised in the Discussion and Conclusion section.

**Summary Of The Paper:**

This paper empirically studies the phenomenon of memorization in deep neural networks for image classification tasks, and presents several novel findings: 1) appropriate data augmentation significantly improves the performance of embedding learning when training with random labels; 2) with data augmentation, there is a clear separation between the final embedding layer applied with $k$-NN and the projection layer -- the embedding is successful at generalizing to true labels, while the projection layer memorizes the random training labels; 3) the benefits of data augmentation in this setting are not explained by label invariance; 4) a possible explanation of the discovered effect is due to augmented datasets increasing the effective size of the dataset beyond the capacity of function class.

**Summary Of The Review:**

Overall, the paper described an interesting empirical investigation into the phenomenon of memorization in noisy label vision classification tasks and came away with several interesting nuggets of insight. I thought the insights were quite interesting and novel, and the paper was for the most part pretty clear to read. I also thought the experiments well supported the claims and made sense. The paper will certainly give theorists more food for thought when thinking about how to study the problems associated with data augmentation and memorization in deep networks.

---

> ### Author Response · Authors · 2022-11-09
> **Response to Reviewer CAZe**
>
> We thank the reviewer for the very precise and thorough reading of our work. We are very glad to hear that our work provides novel and interesting insights into memorization that can inspire theorists to further study the phenomenon. We will address the questions and comments in the following.
>
> 1. **More architectures.** We want to highlight that we have verified our observations on two architectures, ResNets and VGG. We are currently running more experiments on additional architectures to further strengthen our results. We will post an update once those experiments are completed.
> 2. **Capacity definition.** We agree with the reviewer that the exact mathematical definition for capacity is not essential to our work. However, we believe that the notion of capacity of a model has been used in a very ambiguous and inconsistent way in the literature and hence wanted to provide a concrete definition.
> 3. **Probing.** We refer the reviewer to the comment addressed to all the reviewers. We find only a marginal difference between $K$-NN and linear probing. By complexity of the probing, we were referring to the fact that the employed models ($K$-NN and linear) themselves cannot perform additional feature learning on top, i.e. they are restricted to “work with what they get” and in that sense have low complexity. We believe that such a probing strategy is most sensible to assess the quality of embeddings, and has been similarly done in a lot of previous works. We have updated the text to better explain this.
> 4. **Invariance measure.** Thank you for this question, we are indeed averaging over multiple datapoints, thus avoiding scaling problems as you correctly point out. Our measure is inspired by the line of works on contrastive self-supervised learning, that use very similar normalized measures as loss functions (e.g. [1]). We have included more details as a new section in the appendix (section D2). Specifically we use a normalization term to ensure that the collapsed solution does not get a high invariance measure score.
> 5. **Citations.** We thank the reviewer for pointing us to these additional works. We have incorporated them in the related works and discussion sections.
> 6. **Data splits.** We have mentioned the splits for the $K$-NN probing in Section 3 in paragraph *Random labels and memorization*. We are happy to further clarify and discuss the setup if it remains unclear for the reviewer.
> 7. **Type of augmentations.** Thank you for pointing this out, we were indeed not clearly stating this. We have now highlighted in the text that we only use i.i.d. augmentations in Section 5 in paragraph *Label preservation*. In any other place, augmentation refers to the standard set of augmentations highlighted in Section 4 in paragraph *Training Details*.
> 8. **Toy example.** Thank you for pointing out that more information for the toy example is needed to eliminate confusion. We have updated the text to better highlight our point. The motivation behind the toy example, comes from trying to define a model for which we can calculate an upper bound for the capacity of the model as defined in Definition 5.1.
>
> [1] Chen, Ting, et al. "A simple framework for contrastive learning of visual representations." International conference on machine learning. PMLR, 2020.

---

> > ### Comment · Reviewer_CAZe · 2022-11-17
> > **Response to Author Response**
> >
> > Thanks for the response!
> >
> > 1. Great to see the update, thanks for adding the extra experiments
> > 2. I would suggest deferring the capacity definition to the appendix and just include a pointer in the paper to it, it is good to clarify what is meant by "capacity" but I think it hurts readability here, especially since the definition being used doesn't add any particular insight as far as I can see.
> > 3. Thanks for the updates on probing and the additional experiments.
> > 4. Great, thanks.
> > 5. Great, thanks.
> > 6. I still didn't see it -- I was referring to the proportions of training data and testing data used in the experiments (i.e. what is the number of examples in the train set and the clean test set).
> > 7. Great, thanks.
> > 8. I still think the choice of projector could still be better motivated, it feels pretty arbitrary to me.
> >
> > I would also appreciate more clarification on the $B \to \infty$ question I had, relating to the "inflated sample size" section on pg. 8. Why is it reasonable to think of $B \to \infty$? I think I understand the point that you're trying to make (large augmentations will create a non-trivial distribution/support which will overwhelm the capacity of the model), but this seems like a difficult argument to make formally (existing capacity bounds in theory are pretty loose typically, and very difficult if you want to consider the set of models reached by GD after some number of steps, as this paper's definition of capacity does), and if it's only applied heuristically, I think it could be phrased bettter without the $B \to \infty$ notation.
> >
> > Otherwise, I think the paper does a great job of investigating and highlighting an interesting phenomenon surrounding data augmentation and the training of deep nets.

---

> > > ### Author Response · Authors · 2022-11-18
> > > **Response to Reviewer**
> > >
> > > We thank the reviewer for his response and the positive conclusion! We answer the remaining concerns in the following:
> > >
> > > **Capacity definition:** We agree that moving the capacity definition to the Appendix is a good compromise and have re-organized the paper accordingly.
> > >
> > > **Data splits:** We apologize for the misunderstanding and have clarified the train and test splits in Appendix F (Table 5).
> > >
> > > **Increasing number of augmentation:** We understand the reviewer’s point, we have replaced the notation in the paper and simply remark that for online augmentations, the number of augmentations increases with training and will at some point exceed the (finite) capacity of the model. We agree that making a formal statement is very difficult due to the complicated nature of deep models.

---

### Author Response · Authors · 2022-11-09
**To All Reviewers**

We thank all the reviewers for their careful reading of our work and the constructive feedback. We are glad to hear that all the reviewers find the topic relevant to the ICLR community. In the following we will address questions shared across reviewers.

**Linear Probing**: We have added the results of linear probing in Table 4 in Appendix A. The results are very similar to $K$-NN probing, which is consistent with the literature (e.g. [1]).

**Releasing Code**: We have released the code to reproduce the main experiments in the supplementary material.

**More architectures**: We remark that we already have evaluated two common architectures showing consistent results. To further strengthen our results, we are currently evaluating more architectures. We will update our rebuttal once the experiments have finished.

[1] Bardes, Adrien, Jean Ponce, and Yann LeCun. "Vicreg: Variance-invariance-covariance regularization for self-supervised learning." arXiv preprint arXiv:2105.04906 (2021).

---

> ### Author Response · Authors · 2022-11-17
> **Update on Experiments**
>
> As promised, to further strengthen our findings, we have evaluated other common vision architectures. We highlight that the main conclusions are consistent across these architectures, also for models that do not use batch normalization, a point raised by reviewer ZaD4.
>
> We extend Table 2 with the $K$-NN probing accuracies of the new architectures, for the *CIFAR-10* dataset.
>
> | MODEL | RANDOM | RANDOM + DA | CLEAN | CLEAN + DA | INIT |
> | :---  |  :----: |  :----: |  :----: |  :----: |  :----: |
> | ResNet with GN | 14.3 | 74.3 | 80.8 | 94.4 | 39.86 |
> | DenseNet | 16,6 | 73.3 | 89.72 | 94.09 | 42.85 |
> | MobileNet | 11.1 | 64.2 | 79.02 | 82.07 | 25.53 |
>
>
> ResNet with GN: ResNet where all batch normalization layers have been replaced with group normalization with $\frac{\tt{channels}}{8}$ number of groups. Here, $\tt{channels}$ is the number of channels in the corresponding layer.

---

> ### Comment · Reviewer_CAZe · 2022-11-17
> **Thanks for the updates!**
>
> Thanks for adding the linear probing experiment and verifying that it does not differ too much from K-NN. It is also great to see the trends extend to a few more architectures.

---

### Decision · Program_Chairs · 2023-01-20

**Decision:**

Accept: poster

**Justification For Why Not Higher Score:**

The work would benefit from more direct comparison to previous self-supervised techniques, which lead to similar conclusions.

**Justification For Why Not Lower Score:**

The authors make in interesting observation that training with random labels, but heavy data augmentation leads models to learn meaningful representations.

**Metareview: Summary, Strengths And Weaknesses:**

This work studies the memorization in neural networks trained on data with random labels. The authors find that without data-augmentation, such training does not lead to meaningful feature learning, however with label-preserving data-augmentations, earlier layers in networks learn meaningful features. These are interesting empirical findings worthy of publication. That being said, I would encourage the authors in the introduction and discussion to more dramatically emphasize the connection to self-supervised techniques (such as SimCLR). In particular, the label-preserving augmentations studied here encourage representations of the same base image with different augmentations to align as much as possible (predict the same random output class label), and to not align with different images (predict different random output vectors). Viewed in this light, one could describe this work as a variation on previous self-supervised approaches. For example, in SimCLR, the lower layers learn meaningful representations which are used for classification after the initial self-supervised training with a projection head used to align images with different augmentations. That being said, it is interesting to see that random labels and heavy augmentations are enough to realize this.

**Note From Pc:**

if the above contains the word "oral" or "spotlight" please see: "oral" presentation means -> notable-top-5% and "spotlight" means -> notable-top-25%. As stated in our emails, we are disassociating presentation type from AC recommendations